# TOPOFORMER: TOPOLOGY MEETS ATTENTION FOR GRAPH LEARNING

**Md Joshem Uddin** [*]
Department of Mathematical Science
The University of Texas at Dallas
Richardson, TX 75080, USA
mdjoshem.uddin@utdallas.edu

**Astrit Tola** [*]
Department of Mathematics
Florida State University
Tallahassee, FL 32306, USA
atola@fsu.edu

**Cuneyt Gurcan Akcora**
AI Initiative
University of Central Florida
Orlando, FL 32816, USA
cuneyt.akcora@ucf.edu

**Baris Coskunuzer**
Department of Mathematical Science
The University of Texas at Dallas
Richardson, TX 75080, USA
coskunuz@utdallas.edu

## ABSTRACT

We introduce TOPOFORMER, a lightweight and scalable framework for graph representation learning that encodes topological structure into attention-friendly sequences. At the core of our method is *Topo-Scan*, a novel module that decomposes a graph into a short, ordered sequence of topological tokens by slicing over node or edge filtrations. These sequences capture multi-scale structural patterns, from local motifs to global organization, and are processed by a Transformer to produce expressive graph-level embeddings. Unlike traditional persistent homology pipelines, *Topo-Scan* is parallelizable, avoids costly diagram computations, and integrates seamlessly with standard deep learning architectures. We provide theoretical guarantees on the stability of our topological encodings and demonstrate state-of-the-art performance across graph classification and molecular property prediction benchmarks. Our results show that TOPOFORMER matches or exceeds strong GNN and topology-based baselines while offering predictable and efficient compute. This work opens a new path for parallelizable and unifying approaches to graph representation learning that integrate topological inductive biases into attention frameworks.

**Code:** https://github.com/joshem163/TOPOFORMER

## 1 INTRODUCTION

Graphs are powerful data structures for modeling relational data in biology, chemistry, and social networks. While recent advances in graph learning have produced strong task-specific models, most architectures lack the generalization of foundation models in vision and language (Radford et al., 2021; Bubeck et al., 2023). Achieving such general-purpose capability in graphs is difficult due to their irregular, non-Euclidean structure (Wu et al., 2020), which complicates the design of transferable inductive biases.

Topological Data Analysis (TDA) provides a principled approach by encoding global and local structure in a way that is stable to perturbations and insensitive to node identity (Hensel et al., 2021; Pham et al., 2025). In principle, Persistent Homology (PH) offers a canonical summary of how connectivity and cycles evolve across scales, and has proven useful across domains (Skaf et al., 2022; Obayashi et al., 2022; Shultz, 2023). In practice, however, PH pipelines depend on persistence diagrams, which require expensive global reductions and a subsequent vectorization step (e.g., images, landscapes, curves) whose design can materially affect downstream performance. On graphs, common

---

[*]Equal contribution.

sublevel/superlevel filtrations also tend to *early-saturate*, high-valued vertices activate early, quickly filling the complex and suppressing late-emerging features. These computational and modeling frictions have slowed the adoption of PH in graph representation learning, despite the clear promise of topological signals for multi-resolution structure.

To overcome these barriers, we develop a lightweight yet expressive alternative that bypasses full persistence diagrams while retaining multi-resolution topological information in a form consumable by transformers. We introduce TOPOFORMER, a scalable framework that integrates topological descriptors with attention architectures. At its core is Topo-Scan, a module that converts a graph into a short, ordered sequence of topological tokens across multiple resolutions. These sequences are directly consumable by attention mechanisms (Vaswani et al., 2017), enabling efficient graph-level representations within the same token-based interface used by large-scale transformer models. We therefore view TOPOFORMER as a step toward topology-aware graph foundation models, rather than a full foundation model itself, and leave large-scale pretraining on heterogeneous graph corpora to future work. TOPOFORMER achieves strong performance on graph classification and molecular property prediction under unified evaluation protocols, with theoretical guarantees on the stability of its topological encodings. Our Contributions are as follows:

- We introduce a scalable method for turning topological structure into attention-ready sequences, enabling transformers to process graphs without relying on node embeddings or heavy preprocessing.

- We propose a new framework that bridges topological data analysis and deep learning, capturing both local and global graph structure through a unified attention mechanism.

- We conduct a comprehensive evaluation across diverse graph learning tasks, demonstrating strong performance on both graph classification and molecular property prediction benchmarks.

- We provide theoretical guarantees on the robustness of our representations and show that our approach offers predictable and efficient compute, making it practical for large-scale applications.

## 2 MOTIVATION AND BACKGROUND

This section reviews recent work and highlights the need to integrate advanced topological methods with modern ML to overcome limitations in graph representation learning.

**Persistent Homology for Graphs.** Persistent Homology (PH) was first defined for filtered simplicial complexes in the early 2000s (Edelsbrunner et al., 2002; Zomorodian & Carlsson, 2005). Early applications centered on point clouds $\mathcal{X} \subset \mathbb{R}^N$, where Vietoris–Rips filtrations generate nested complexes $\Delta_1(\mathcal{X}) \subset \Delta_2(\mathcal{X}) \subset \cdots$, allowing topological features to be tracked across scales (Carlsson, 2009). The persistence diagram $\mathrm{PD}_k(\mathcal{X}) = \{[b_i, d_i]\}$ records births and deaths of $k$-dimensional features, with longer intervals $(d_i - b_i)$ interpreted as more persistent and thus more structurally significant (Dey & Wang, 2022).

PH has since been applied to graphs and images. For graphs, two principal approaches are used. *Power (distance) filtrations* treat nodes as a point cloud with graph distances as pairwise distances, then build a Rips filtration (Aktas et al., 2019), which is typically computationally heavy. A more practical alternative in graph learning is *sublevel filtrations*, where a scalar node/edge function $f$ induces nested subgraphs that are lifted to simplicial complexes via cliques (upper–star extension is standard). A key interpretability difference follows: in power filtrations, bar lengths reflect geometric scale; in sublevel filtrations, they reflect *differences in $f$* rather than physical size, so "long bars $\Rightarrow$ important features" need not hold universally. Poorly chosen $f$ may yield many short-lived features or early saturation, while task-aligned or *learnable* filtrations can mitigate these effects (Hofer et al., 2020). Rather than viewing this as an intrinsic weakness, we take it as motivation to design fixed-budget, stable summaries that integrate smoothly with modern ML (See App. C.8 for discussion).

The standard PH pipeline for graphs has three main steps (Coskunuzer & Akçora, 2024): *filtration, persistence computation*, and *vectorization*. Given a graph $\mathcal{G} = (\mathcal{V}, \mathcal{E})$, a function $f : \mathcal{V} \to \mathbb{R}$ with thresholds $\{\alpha_i\}_{i=1}^N$ induces subgraphs $\mathcal{G}_1 \subseteq \cdots \subseteq \mathcal{G}_N$, where $\mathcal{G}_i$ contains vertices $\{v \in \mathcal{V} \mid f(v) \leq \alpha_i\}$. Lifting each $\mathcal{G}_i$ to its clique complex $\widehat{\mathcal{G}}_i$ yields a filtration $\{\widehat{\mathcal{G}}_i\}$. Persistence diagrams $\mathrm{PD}_k(\mathcal{G}, f) = \{(b_j, d_j)\}$ record births and deaths of $H_k(\widehat{\mathcal{G}}_i)$ and are typically vectorized via persistence images, landscapes, or Betti curves (Ali et al., 2022).

In recent years, the ML community has increasingly recognized the value of topological encodings for graph-level tasks, with PH-based methods showing strong results across domains (Immonen et al., 2023; Demir et al., 2022; Verma et al., 2024; Loiseaux et al., 2023b; Horn et al., 2021; Chen et al., 2024a). Despite this promise, two bottlenecks hinder broader adoption: (1) the computational overhead of persistence computations in large pipelines (Otter et al., 2017), and (2) the difficulty of choosing vectorizations that align with downstream objectives (Ali et al., 2022). Our TOPOFORMER framework addresses both by producing a compact *sequence* of stable, low-cost topological tokens that feed directly into attention layers, thereby bypassing full persistence diagrams and bespoke vectorizations while remaining compatible with efficient graph-specific computations.

Recent methods learn neural approximations of persistence-based topological features to reduce the cost of exact PH. RipsNet (de Surrel et al., 2022) estimates Rips persistence diagrams for point clouds directly from raw data, while Yan et al. (2022) approximate graph topological features with a GNN. Our approach is complementary: instead of approximating persistence diagrams, Topo-Scan bypasses global PH and directly builds short interlevel topological sequences tailored to Transformer encoders.

**Transformers.** Transformers (Vaswani et al., 2017) underpin transferable models in language and vision (Devlin, 2018; Dosovitskiy et al., 2020) by learning from *ordered* token sequences with long-range dependencies. On graphs, adapting attention is challenging due to variable size, permutation invariance, and irregular connectivity. Our design sidesteps these issues: Topo-Scan yields a short, 1D *ordered* sequence of topological tokens with a fixed channel width, so positional encodings and attention operate in their native regime, without graph-specific heavy machinery. This makes transformers a natural, efficient backend for multi-resolution structural signals. By contrast, recent graph transformer models such as Graphormer (Ying et al., 2021), GPS (Rampášek et al., 2022), and related architectures operate directly on node tokens and inject structure via shortest-path or Laplacian-based positional encodings and attention biases. In TOPOFORMER, each graph is first compressed into a short sequence of topological tokens, so attention runs on a fixed-length, purely topological sequence rather than on all nodes of the original graph.

**Molecular Property Prediction.** Molecular property prediction (MPP) is central to drug discovery (ADMET). Classical pipelines use engineered fingerprints with RF/SVMs (Cereto-Massagué et al., 2015); deep learning extends to MLPs on fingerprints, sequence models on SMILES (Rong et al., 2020), and GNNs on molecular graphs (Wieder et al., 2020), with recent 3D methods trading accuracy for higher compute and sensitivity to rotations (Gasteiger et al., 2021; Li et al., 2022b). Despite progress, DL does not always surpass strong classical baselines on realistic benchmarks (Janela & Bajorath, 2022; Valsecchi et al., 2022), motivating transformer variants (Sultan et al., 2024), geometric models (Liu et al., 2022b), and topological approaches (Demir et al., 2022; Loiseaux et al., 2023a). Evaluation protocols also vary (e.g., scaffold vs. random splits), affecting reported generalization. Our approach unifies robust topological structure with a scalable attention backend, providing an effective, split-agnostic representation for MPP.

## 3  TOPO-TRANSFORMERS

TDA captures multi-scale structural patterns while offering robustness to noise, making it attractive for representation learning. However, the standard Persistent Homology pipeline, consisting of filtration construction, persistence diagram computation, and vectorization, introduces inefficiencies and lacks adaptability, particularly in graph settings. While persistence diagram computation is standardized, vectorization remains ad hoc and significantly impacts model performance (Ali et al., 2022). Our goal is to develop an efficient and scalable alternative to this workflow for graph representation learning.

Our first insight is that the *strict nestedness condition required in PH is not always necessary for graphs*. Unlike point clouds, graphs inherently encode structural relationships that permit more flexible and direct

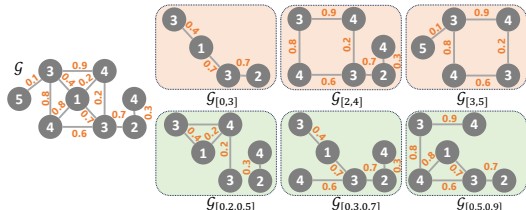

Figure 1: **Topo-Scan**. Topo-Scan decomposes a graph into sequential topological slices via node and edge filtrations. The top row shows node-based filtrations; the bottom row, edge-based ones.

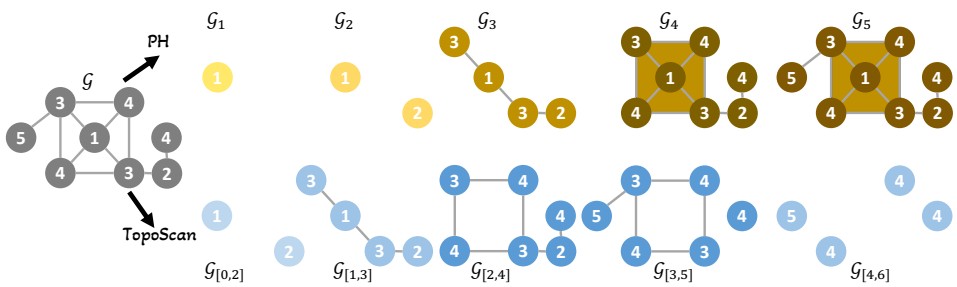

Figure 3: **Topo-Scan vs. PH.** This toy example highlights the key differences between the Topo-Scan filtration and standard PH filtration, where node values indicate filtration function values. In PH, early-activated nodes quickly saturate the graph, suppressing the emergence of later topological features. As shown, PH yields relatively uninformative barcodes with $\beta_0 = \langle 1, 2, 1, 1, 1 \rangle$ and $\beta_1 = \langle 0, 0, 0, 0, 0 \rangle$, while Topo-Scan captures richer topological dynamics, producing $\beta_0 = \langle 2, 1, 1, 2, 4 \rangle$ and $\beta_1 = \langle 0, 0, 1, 1, 0 \rangle$.

extraction of topological features. Building on this observation, we bypass persistence diagrams and vectorization by directly extracting topological sequences from structured graph slices. This shift enables efficient and adaptable pattern extraction and forms the foundation of a scalable learning framework.

**Topo-Scan.** Traditional sublevel filtrations on graphs often saturate rapidly, and once most nodes join the subgraph at low thresholds, little new structure emerges and important patterns at larger scales are lost. Topo-Scan overcomes this by first imposing a directional hierarchy via a scalar function $f : \mathcal{V} \to \mathbb{R}$, then slicing the graph into a sequence of overlapping subgraphs along increasing values of $f$. Rather than waiting for a single threshold to engulf the entire graph, each slice captures fresh topological information, such as connectivity changes and emerging loops, without early collapse. We compute basic invariants (e.g. Betti numbers) on each slice to form a compact, ordered signature sequence. Feeding these ordered descriptors into a transformer lets the model attend to structure at every scale, ensuring no signal is lost to premature saturation (See Fig. 3 for a toy example).

Let $f : \mathcal{V} \to \mathbb{R}$ be a filtration function defined on the vertices, with thresholds $\alpha_0 = \min_{v \in \mathcal{V}} f(v) < \alpha_1 < \cdots < \alpha_N = \max_{v \in \mathcal{V}} f(v)$. In most cases, the thresholds are selected either as evenly spaced values or based on quintiles. Next, for each $\alpha_i$, we define $\mathcal{V}_i = \{v_r \in \mathcal{V} \mid \alpha_i \le f(v_r) \le \alpha_{i+m}\}$ and $\mathcal{G}_i$ as the induced subgraph $\mathcal{G}_i = (\mathcal{V}_i, \mathcal{E}_i)$, where $\mathcal{E}_i = \{e_{rs} \in \mathcal{E} \mid v_r, v_s \in \mathcal{V}_i\}$. The clique complex of $\mathcal{G}_i$, denoted $\widehat{\mathcal{G}}_i$, forms a sequence $\{\widehat{\mathcal{G}}_i\}$ called *slicing*. We call this process *Topo-Scan*, which decomposes graphs into topological slices, similar to medical scans revealing structural layers. Leveraging a hierarchical structure, it adapts to node and edge filtrations, weighted graphs, and diverse relations, capturing local and global topological patterns for robust, scalable representation learning. It remains robust by tracking short-lived features effectively and is scalable through parallelized slice extraction.

The resolution ($N$) determines the number of slices, while the thickness ($m$) specifies the range of nodes included in each slice. After constructing $\{\widehat{\mathcal{G}}_i\}$, we compute four outputs for each slice: $\beta_0(\widehat{\mathcal{G}}_i)$ (Betti-0, connected components), $\beta_1(\widehat{\mathcal{G}}_i)$ (Betti-1, cycles/holes), $|\mathcal{V}_i|$ (node count), and $|\mathcal{E}_i|$ (edge count). These outputs form ordered sequences of size $N$, such as $\widehat{\beta}_k(\mathcal{G}) = \{\beta_k(\widehat{\mathcal{G}}_i)\}_{i=1}^N$ for $k = 0, 1$. While $\widehat{\beta}_k(\mathcal{G})$ are the primary topological outputs, $\{|\mathcal{V}_i|\}$ and $\{|\mathcal{E}_i|\}$ serve as normalization factors (see Figure 1). These sequences are concatenated into a sequence (vector) $\Gamma(\mathcal{G})$ of length $4N$ where $N$ is the number of slices.

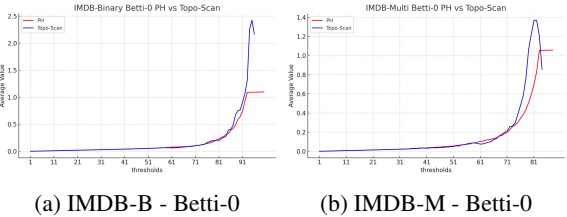

(a) IMDB-B - Betti-0  (b) IMDB-M - Betti-0

Figure 2: **PH vs. Topo-Scan.** Average Betti-0 counts under degree filtration with 100 thresholds on IMDB-B and IMDB-M. Standard PH saturates early, causing a sharp decline near the end of the curve. Topo-Scan avoids early saturation and continues to reveal late-emerging features, often surpassing PH counts at higher thresholds.

A key distinction from PH lies in activation: PH includes all nodes up to a threshold, causing early saturation in dense graphs, while Topo-Scan uses a sliding window to preserve late-emerging features

and capture fine structure (see Fig. 2 and App. C.8). Slice thickness $m$ controls locality, allowing flexibility across datasets. Its localized design ensures robustness to noise and enables parallelization, making it ideal for scalable ML workflows.

*Vectorization Choice.* Among many possible vectorizations, we deliberately use a very low-dimensional token per slice, $(\beta_0, \beta_1, |\mathcal{V}_i|, |\mathcal{E}_i|)$. Global vectorizations such as persistence images or landscapes aggregate information over the entire filtration into a single feature vector, which largely destroys the *sequential* structure that Topo-Scan is designed to expose. In contrast, our Betti-based tokens preserve how components and cycles evolve across slices; richer per-slice invariants could be plugged in, but we focus on this minimal choice to isolate the benefit of the sequential representation.

**TOPOFORMER.** We use the *ordered sequences* of topological features in Transformers, which excel at capturing sequential structures and complex dependencies through self-attention mechanisms, making them well-suited for tasks requiring order and contextual understanding. While traditional PH processes a sequence of simplicial complexes $\{\Delta_i\}$, this sequential structure is often lost during the persistence diagram and vectorization stages, where outputs are transformed into unordered vectors. Topo-Scan preserves the sequential nature of topological features and aligns them with transformers' ability to model positional relationships.

**ML Model.** Our transformer architecture (Fig. 4) consists of an embedding layer that processes input sequences, a transformer encoder that captures hierarchical dependencies through self-attention mechanisms, and a fully connected classification head that maps learned representations to output predictions. To enhance generalization and mitigate overfitting, we integrate regularization techniques, such as dropout and weight decay, ensuring robustness across diverse graph learning tasks. Formally, given an input sequence $\mathbf{x} \in \mathbb{R}^{m \times T \times D}$, where $m$ is the number of graphs, $T$ the sequence length, and $D$ the token dimensionality, the sequence is embedded via $\mathbf{E}$, with positional encoding $\mathbf{P}$ added. This processed sequence is then passed through a multi-layer transformer encoder, producing an output representation $\mathbf{z}$, which is flattened and normalized before being classified via a fully connected layer.

Expanding this model, we introduce a dual-transformer framework with an integrated multi-layer perceptron (MLP) classifier to handle diverse input modalities. The model processes three distinct inputs: $\mathbf{X}_1$ and $\mathbf{X}_2$ through independent transformers $\mathcal{T}_1$ and $\mathcal{T}_2$, and $\mathbf{X}_3$ through an MLP $\mathcal{M}$. Their respective outputs $\mathbf{z}_1$, $\mathbf{z}_2$, and $\mathbf{z}_3$ are combined using a learnable weighted sum, allowing the model to adaptively balance feature contributions:

$$\mathbf{z}_{\text{combined}} = \alpha \cdot \mathbf{z}_1 + \beta \cdot \mathbf{z}_2 + (1 - \alpha - \beta) \cdot \mathbf{z}_3$$

where $\alpha$ and $\beta$ are learned during training. The aggregated feature vector is then batch-normalized and passed through a fully connected layer to produce the final classification output: $\hat{y} = \mathbf{FC}(\mathbf{z}_{\text{combined}})$ where $\hat{y} \in \mathbb{R}^C$ represents the class probabilities, with $C$ being the number of output classes. Model details are given in Appendix C.

### 3.1 STABILITY OF TOPO-SCAN SEQUENCES

A useful graph vectorization should be robust: small changes in the filtration signal should not cause large changes in the output sequence. We formalize this for Topo-Scan on the *fixed clique complex* $\widehat{G}$ of $G = (V, E)$ using upper–star extensions of node functions.

**Setup.** Let $f, g : V \to \mathbb{R}$ be filtration functions, extended to $\widehat{G}$ by the upper–star rule $\widehat{h}(\sigma) = \max_{v \in \sigma} h(v)$. Fix a shared threshold grid $\alpha_0 < \cdots < \alpha_N$, window width $m$, stride $s$, and windows $I_t = [\alpha_{ts}, \alpha_{ts+m}]$ for $t = 0, \ldots, T-1$, where $T = \lfloor (N - m)/s \rfloor + 1$. For $k \in \{0, 1\}$, the $t$-th Topo-Scan token is the interlevel Betti number

$$\widehat{\beta}_k^h(t) := \dim H_k\big((\widehat{G})_{I_t}^h\big), \qquad h \in \{f, g\}.$$

**Theorem 3.1** (Discrete $\ell_1$ stability of Topo-Scan). *There exists $C = C(\widehat{G}, \{\alpha_i\}, m, s)$ such that for $k \in \{0, 1\}$,*

$$\big\| \widehat{\beta}_k(G, f) - \widehat{\beta}_k(G, g) \big\|_1 \leq C \ d_B(M_k^f, M_k^g),$$

*where $M_k^h$ denotes the $k$-dimensional interlevel (level-set) persistence module of the upper–star filtration induced by $h$ on $\widehat{G}$, and $d_B$ is the bottleneck distance between such modules.*

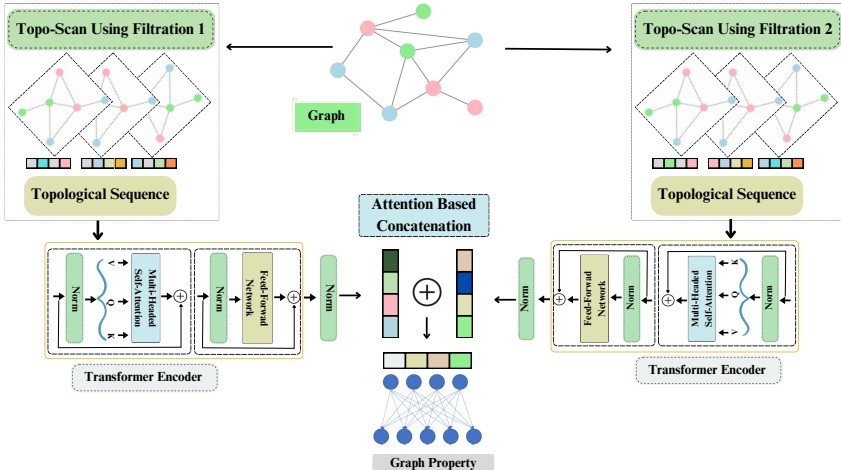

Figure 4: **TopoFormer Flowchart**: Given an input graph $\mathcal{G}$, sequential substructures are extracted via Topo-Scan. Each substructure is encoded into a four-dimensional topological signature. These sequences are processed by a transformer model, and outputs from multiple filtration functions are fused using attention-based concatenation. A final prediction layer maps the representation to the target graph property.

**Corollary 3.2.** *For upper–star filtrations on a fixed complex, interlevel modules satisfy* $d_B(M_k^f, M_k^g) \leq \|f - g\|_\infty$. *Hence* $\|\widehat{\beta}_k(G, f) - \widehat{\beta}_k(G, g)\|_1 \leq C \|f - g\|_\infty$.

**Outline.** Each token counts classes surviving exactly over $I_t$; under a $\delta$ bottleneck matching, only classes within $\delta$ of the interval boundary can change their contribution, so per-window changes are $O(\delta)$ and summing over windows yields the discrete $\ell_1$ bound with a constant depending on the window schedule and the (finite) bar complexity of $\widehat{G}$. Full details and references are in Appendix B.

Beyond stability, the Topo-Scan sequences $\widehat{\beta}_k(G, h)$ are closely related to classical PH invariants: they can be viewed as a discrete sampling of the rank invariant / Betti curve of the interlevel module $M_k^h$ along our window schedule. Thus, Topo-Scan provides a coarse but structured, Transformer-ready discretization of the same homological information underlying barcodes and stable-rank summaries; see Appendix Remark B.3 for further discussion.

## 4 EXPERIMENTS

### 4.1 EXPERIMENTAL SETUP

**Datasets.** We report the TOPOFORMER performance in two graph learning tasks: **graph classification** and **molecular property prediction (MPP)**.

*Graph Classification Datasets.* We use nine graph classification benchmark datasets: (i) molecular graphs from BZR, MUTAG and COX2 (Kriege & Mutzel, 2012); (ii) biological graphs PROTEINS (Borgwardt et al., 2005); and (iii) social graphs, including IMDB-Binary, IMDB-Multi, REDDIT-Binary, and REDDIT-Multi-5K (Yanardag et al., 2015). We also include a large-scale dataset, OGBG-MOLHIV, from Open Graph Benchmark (Hu et al., 2020b). Dataset statistics are provided in Table 1.

*MPP Datasets.* For molecular property prediction (MPP), we employ seven datasets from MoleculeNet (Wu et al., 2018): BBBP (blood-brain barrier penetration), Tox21, ToxCast, ClinTox (toxicity

Table 1: Graph classification datasets.

| Datasets | #Graphs | $|\mathcal{V}|$ | $|\mathcal{E}|$ | Classes |
|---|---|---|---|---|
| BZR | 405 | 35.75 | 38.36 | 2 |
| COX2 | 467 | 41.22 | 43.45 | 2 |
| MUTAG | 188 | 17.93 | 19.79 | 2 |
| PROTEINS | 1113 | 39.06 | 72.82 | 2 |
| IMDB-B | 1000 | 19.77 | 96.53 | 2 |
| IMDB-M | 1500 | 13.00 | 65.94 | 3 |
| REDDIT-B | 2000 | 429.63 | 497.75 | 2 |
| REDDIT-5K | 4999 | 508.52 | 594.87 | 5 |
| OGBG-MOLHIV | 41127 | 25.5 | 27.5 | 2 |

Table 2: **Graph Classification.** Accuracy on eight benchmark datasets using 10-fold CV. Baseline results are taken from the respective papers using the same setting. We mark the 1st (blue), 2nd (purple), and 3rd (green) per column. The last two columns report the average deviation (AvD) from the best-performing model and the average rank (AvR) across all datasets.

| Model | BZR | COX2 | MUTAG | PROTEINS | IMDB-B | IMDB-M | REDDIT-B | REDDIT-5K | AvD↓ | AvR↓ |
|---|---|---|---|---|---|---|---|---|---|---|
| 6 GNNs (Errica et al., 2020) | – | – | $80.42_{\pm2.07}$ | $75.80_{\pm3.70}$ | $71.20_{\pm3.90}$ | $49.10_{\pm3.50}$ | $89.90_{\pm1.90}$ | $56.10_{\pm1.60}$ | 6.0 | 8.5 |
| PersLay (Carrière et al., 2020) | – | $80.90_{\pm\text{NA}}$ | $89.80_{\pm\text{NA}}$ | $74.80_{\pm\text{NA}}$ | $71.20_{\pm\text{NA}}$ | $48.80_{\pm\text{NA}}$ | – | $55.60_{\pm\text{NA}}$ | 5.2 | 8.7 |
| DMP (Bodnar et al., 2021) | – | – | $84.00_{\pm8.60}$ | $75.30_{\pm3.30}$ | $73.80_{\pm4.50}$ | $50.90_{\pm2.50}$ | $86.20_{\pm6.80}$ | $51.90_{\pm2.10}$ | 6.1 | 8.3 |
| FC-V (O'Bray et al., 2021) | $85.61_{\pm0.59}$ | $81.01_{\pm0.88}$ | $87.31_{\pm0.66}$ | $74.54_{\pm0.48}$ | $73.84_{\pm0.36}$ | $46.80_{\pm0.37}$ | $89.41_{\pm0.24}$ | $52.36_{\pm0.37}$ | 5.7 | 9.2 |
| SubMix (Yoo et al., 2022) | $86.34_{\pm2.00}$ | $84.68_{\pm3.70}$ | $80.99_{\pm0.60}$ | $67.80_{\pm2.00}$ | $70.30_{\pm1.40}$ | $46.47_{\pm2.50}$ | – | – | 8.4 | 11.5 |
| G-Mix (Han et al., 2022) | $84.15_{\pm2.30}$ | $83.83_{\pm2.10}$ | $81.96_{\pm0.60}$ | $66.28_{\pm1.10}$ | $69.40_{\pm1.10}$ | $46.40_{\pm2.70}$ | – | – | 9.1 | 12.8 |
| RGCL (Li et al., 2022a) | $84.54_{\pm1.67}$ | $79.31_{\pm0.68}$ | $87.66_{\pm1.01}$ | $75.03_{\pm0.43}$ | $71.85_{\pm0.84}$ | $49.31_{\pm0.42}$ | $90.34_{\pm0.58}$ | $56.38_{\pm0.40}$ | 5.2 | 8.0 |
| AutoGCL (Yin et al., 2022) | $86.27_{\pm0.71}$ | $79.31_{\pm0.70}$ | $88.64_{\pm1.08}$ | $75.80_{\pm0.36}$ | $72.32_{\pm0.93}$ | $50.60_{\pm0.80}$ | $88.58_{\pm1.49}$ | $56.75_{\pm0.18}$ | 4.7 | 7.0 |
| WWLS (Fang et al., 2023b) | $88.02_{\pm0.61}$ | $81.58_{\pm0.91}$ | $88.30_{\pm1.23}$ | $75.35_{\pm0.74}$ | $75.08_{\pm0.31}$ | $51.61_{\pm0.62}$ | – | – | 4.5 | 5.2 |
| PGOT (Qian et al., 2024) | $87.32_{\pm3.90}$ | $82.98_{\pm5.21}$ | $92.63_{\pm2.58}$ | $73.21_{\pm2.59}$ | $62.90_{\pm3.05}$ | $51.33_{\pm1.76}$ | – | – | 6.1 | 7.5 |
| EMP (Chen et al., 2024b) | – | – | $88.79_{\pm0.63}$ | $72.78_{\pm0.54}$ | $74.44_{\pm0.45}$ | $48.01_{\pm0.42}$ | $91.03_{\pm0.22}$ | $54.41_{\pm0.32}$ | 4.8 | 7.5 |
| EPIC (Heo et al., 2024) | $88.78_{\pm2.30}$ | $85.53_{\pm1.60}$ | $82.44_{\pm0.70}$ | $69.06_{\pm1.00}$ | $71.70_{\pm1.00}$ | $47.93_{\pm1.30}$ | – | – | 6.9 | 9.0 |
| MP-HSM (Loiseaux et al., 2023b) | – | $77.10_{\pm3.00}$ | $85.60_{\pm5.30}$ | $74.60_{\pm2.10}$ | $74.80_{\pm2.50}$ | $47.90_{\pm3.20}$ | – | – | 6.9 | 10.1 |
| TopoGCL (Chen et al., 2024a) | $87.17_{\pm0.83}$ | $81.45_{\pm0.55}$ | $90.09_{\pm0.93}$ | $77.30_{\pm0.89}$ | $74.67_{\pm0.32}$ | $52.81_{\pm0.31}$ | $90.40_{\pm0.53}$ | – | 3.5 | 4.3 |
| DASP (Ye et al., 2025) | $89.40_{\pm3.10}$ | $84.80_{\pm4.60}$ | $91.90_{\pm8.60}$ | $77.20_{\pm3.10}$ | $81.40_{\pm3.60}$ | $51.20_{\pm2.20}$ | – | $57.60_{\pm1.60}$ | 1.6 | 2.8 |
| TOPOFORMER | $92.36_{\pm4.11}$ | $83.93_{\pm4.03}$ | $94.68_{\pm4.30}$ | $77.64_{\pm3.64}$ | $78.90_{\pm3.31}$ | $55.40_{\pm4.78}$ | $91.50_{\pm1.89}$ | $57.99_{\pm1.94}$ | 0.5 | 1.5 |

prediction), SIDER (adverse drug reactions), HIV (replication inhibition), and BACE ($\beta$-secretase 1 inhibitors). Dataset statistics are provided in Table 3 (top rows).

**Model Setup.** We use *Topo-Scan* to generate topological signature sequences, which are fed to Transformer classifiers. Each filtration (20 thresholds, width 2) yields four sequences of length 19 (Betti-0, Betti-1, node count, edge count), giving 76 features per filtration. For graph classification, we use Ollivier–Ricci curvature and Heat Kernel Signature; and for molecular property prediction including the OGBG-MOLHIV dataset, we use atomic weight and Ollivier–Ricci curvature. Independent Transformers process each filtration, and their outputs are fused by attention before a final linear layer.

For MPP, we use TOPOFORMER with the standard molecular fingerprints, processed by a two-layer MLP and combined with topological features via attention, yielding TOPOFORMER*. We report 10-fold CV accuracy on graph classification, scaffold-split AUCs over three runs for MPP (Fang et al., 2023a), and use the standard split for OGBG-MOLHIV.

**Hyperparameters.** For model optimization, we employed the Adam optimizer with a learning rate of 0.001. We also use standard regularization techniques—dropout (0.5), weight decay (1e-4), and batch normalization—commonly employed in transformer training. The transformer model architecture was designed with a hidden dimension of 32. Hyperparameters are given in App. C.5.

**Computational Complexity.** Classical PH requires global boundary–matrix reductions with cubic worst-case cost and poor parallelism (Otter et al., 2017). TOPOFORMER *skips persistence diagrams entirely*: instead of global reductions, it computes $\beta_0$ and $\beta_1$ *per slice* on the clique-complex 2-skeleton. $\beta_0$ uses union–find on the 1-skeleton, while $\beta_1$ is derived from sparse edge–triangle operators after triangle enumeration (no cycle-rank identity due to clique complexes). This yields $\mathcal{O}(|V_t| + |E_t| + T_t)$ per slice, aggregated as $\mathcal{O}(L\sum_t(|V_t| + |E_t| + T_t))$ across $k$ slices and $L$ filtrations. Since slices are independent, Betti computations are fully parallelizable. By bypassing PD computation and vectorization, TOPOFORMER achieves multi-fold runtime and memory gains while retaining task-relevant topological features (Appendix C.3). In Appendix C.6, we further show that TOPOFORMER consistently outperforms classical PH across multiple filtration functions and vectorization schemes in the graph classification task.

**Implementation and Runtime.** We implemented our approach in Python and conducted experiments on a 12th Gen Intel Core i7-1270P vPro processor (E-cores up to 3.50 GHz, P-cores up to 4.80 GHz) with 32GB LPDDR5-6400MHz RAM. Topo-Scan feature extraction took 269.38 seconds for OGBG-MOLHIV/HIV and 29.51 seconds for REDDIT-5K; other datasets were faster. The remaining model runtime was negligible. More timeruns and a comparison with PH can be found at Appendix C.3.

## 4.2 RESULTS

**Graph Classification Baselines.** We evaluate our method against 20 state-of-the-art baselines spanning several categories. These include: *GNN-based models* such as GCN, DGCNN, DiffPool,

Table 3: **SOTA MPP Models.** ROC AUC comparison on molecular property prediction with scaffold splitting. We mark the 1st (blue), 2nd (purple), and 3rd (green) per column. The last two columns report the average deviation (AvD) from the best-performing model and the average rank (AvR) across all datasets.

| Model | BBBP | Tox21 | ToxCast | SIDER | ClinTox | BACE | HIV | AvD↓ | AvR↓ |
|---|---|---|---|---|---|---|---|---|---|
| # Molecules | 2,039 | 7,831 | 8,577 | 1,427 | 1,480 | 1,513 | 41,913 | | |
| # Task | 1 | 12 | 617 | 27 | 2 | 1 | 1 | | |
| N-GRAM (Liu et al., 2019) | $91.2_{\pm3.0}$ | $76.1_{\pm2.7}$ | – | $63.2_{\pm0.5}$ | $87.5_{\pm2.7}$ | $79.1_{\pm1.3}$ | $78.7_{\pm0.4}$ | 8.5 | 8.4 |
| PT-GNN (Hu et al., 2020a) | $70.8_{\pm1.5}$ | $78.7_{\pm0.4}$ | $65.7_{\pm0.6}$ | $62.7_{\pm0.8}$ | $72.6_{\pm1.5}$ | $84.5_{\pm0.7}$ | $79.9_{\pm0.7}$ | 12.4 | 8.7 |
| CMPNN (Song et al., 2020) | $92.7_{\pm1.7}$ | $80.3_{\pm1.3}$ | $70.8_{\pm1.3}$ | $61.0_{\pm3.6}$ | $89.8_{\pm0.8}$ | $86.7_{\pm0.2}$ | $78.2_{\pm2.2}$ | 6.1 | 6.2 |
| MGSSL (Zhang et al., 2021) | $70.5_{\pm1.1}$ | $74.0_{\pm1.4}$ | $64.1_{\pm0.7}$ | $59.2_{\pm0.6}$ | $80.7_{\pm2.1}$ | $79.7_{\pm0.8}$ | $79.5_{\pm1.1}$ | 13.5 | 11.7 |
| GEM (Fang et al., 2022) | $70.5_{\pm2.0}$ | $78.1_{\pm0.6}$ | $68.6_{\pm0.2}$ | $63.2_{\pm1.5}$ | $90.3_{\pm0.7}$ | $87.9_{\pm1.0}$ | $81.3_{\pm0.3}$ | 8.9 | 6.6 |
| GROVER (Rong et al., 2020) | $86.8_{\pm2.2}$ | $82.0_{\pm1.6}$ | $56.8_{\pm3.4}$ | $61.2_{\pm2.5}$ | $70.3_{\pm13.7}$ | $82.8_{\pm3.6}$ | $68.2_{\pm1.1}$ | 13.5 | 9.9 |
| GraphMVP (Liu et al., 2022a) | $72.4_{\pm1.6}$ | $76.5_{\pm0.4}$ | $63.1_{\pm0.4}$ | $63.9_{\pm1.2}$ | $79.1_{\pm2.8}$ | $81.2_{\pm0.9}$ | $77.0_{\pm1.2}$ | 12.7 | 10.4 |
| MolCLR (Wang et al., 2022) | $72.6_{\pm1.3}$ | $77.2_{\pm0.6}$ | $65.9_{\pm2.1}$ | $61.3_{\pm6.6}$ | $89.8_{\pm2.7}$ | $88.5_{\pm2.2}$ | $77.4_{\pm0.6}$ | 9.9 | 7.8 |
| MolCLR-2 (Wang et al., 2022) | $72.4_{\pm0.7}$ | $78.4_{\pm0.6}$ | $69.1_{\pm1.2}$ | $59.7_{\pm3.4}$ | $88.0_{\pm4.0}$ | $85.0_{\pm2.4}$ | $77.8_{\pm5.5}$ | 10.2 | 8.6 |
| KANO (Fang et al., 2023a) | $96.0_{\pm1.6}$ | $83.7_{\pm1.3}$ | $73.2_{\pm1.6}$ | $65.2_{\pm0.8}$ | $94.4_{\pm0.3}$ | $93.1_{\pm2.1}$ | $85.1_{\pm2.2}$ | 1.6 | 2.0 |
| MV-Mol (Luo et al., 2024) | $73.6_{\pm0.2}$ | $80.3_{\pm0.6}$ | $70.0_{\pm0.4}$ | $67.3_{\pm0.0}$ | $95.6_{\pm1.6}$ | $88.2_{\pm0.4}$ | $81.4_{\pm0.3}$ | 6.5 | 3.6 |
| MolFuse (Zheng et al., 2024) | $74.3_{\pm1.3}$ | $77.6_{\pm0.4}$ | $64.1_{\pm0.3}$ | $69.5_{\pm1.0}$ | $95.5_{\pm3.3}$ | $87.2_{\pm1.3}$ | $78.6_{\pm0.9}$ | 7.9 | 6.2 |
| MORE (Son et al., 2025) | $71.9_{\pm0.9}$ | $75.6_{\pm0.5}$ | $64.6_{\pm0.6}$ | $60.9_{\pm0.6}$ | $81.0_{\pm0.7}$ | $82.8_{\pm1.3}$ | $77.0_{\pm0.7}$ | 12.6 | 11.1 |
| TOPOFORMER* | $89.5_{\pm1.3}$ | $82.7_{\pm0.5}$ | $75.3_{\pm0.5}$ | $63.1_{\pm0.7}$ | $96.5_{\pm0.6}$ | $95.9_{\pm0.3}$ | $81.2_{\pm0.8}$ | 2.5 | 2.8 |

ECC, GIN, and GraphSAGE (with the best results reported by Errica et al. (2020)); *topological methods* including PersLay, DMP, FC-V, WWLS, MP-HSM, and EMP; *GNNs with data augmentation* such as SubMix, G-Mix, and EPIC; *contrastive learning methods* including RGCL, AutoGCL, and TopoGCL; and *prototype-based methods* such as PGOT. We further include the recent graph kernel method DASP (Ye et al., 2025). A complete list of baselines is provided in Table 2.

**Graph Classification Results.** In graph classification, TOPOFORMER attains the *best or second-best accuracy on 7 out of 8 benchmarks* (Table 2). Aggregating across datasets, it achieves an *average deviation (AvD)* of *0.5* from the best model and an *average rank (AvR)* of *1.5*, demonstrating consistent top-tier performance. Notably, TOPOFORMER establishes new state-of-the-art on BZR, MUTAG,PROTEINS, IMDB-M, REDDIT-B, and REDDIT-5K, while remaining highly competitive elsewhere. It also surpasses common pooling-based methods on these datasets (see Table 12). On the large-scale OGBG-MOLHIV benchmark (Table 4), TOPOFORMER* reaches an AUC within ∼2 points of the strong Graphormer baseline, underscoring both its scalability and the strength of topological signals as an inductive bias in graph learning. For this table, we restrict baselines to peer-reviewed published methods reported in the literature, rather than including unpublished leaderboard entries in (Hu et al., 2020b).

**MPP Baselines.** We compare against strong supervised, self-supervised, and contrastive methods for molecular property prediction (MPP). *Supervised:* CMPNN (message passing on molecular graphs). *Predictive self-supervision:* N-GRAM, PT-GNN, GROVER, MGSSL, GEM. *Contrastive/augmentation and 3D:* GraphMVP (with 3D), MolCLR, MolCLR-2. *Knowledge-aware / prompts:* KANO.

*Recent multi-view/fusion models:* MV-Mol (multi-view molecular representations), MORE (modality-aware molecular representation learning), and Mol-Fuse (fusion of heterogeneous molecular signals). See Table 3 for full references.

**MPP Results.** On molecular property prediction, TOPOFORMER shows strong adaptability when paired with Extended Connectivity Fingerprints.

Table 4: ROC AUC results for OGBG-MOLHIV dataset.

| Model | ROC AUC |
|---|---|
| GIN-VN (Xu et al., 2018) | $77.80_{\pm1.82}$ |
| HGK-WL (Togninalli et al., 2019) | $79.05_{\pm1.30}$ |
| WWL (Borgwardt et al., 2020) | $75.58_{\pm1.40}$ |
| PNA (Corso et al., 2020) | $79.05_{\pm1.32}$ |
| DGN (Beaini et al., 2021) | $79.70_{\pm0.97}$ |
| GraphSNN (Wijesinghe et al., 2021) | $79.72_{\pm1.83}$ |
| GCN-GNorm (Cai et al., 2021) | $78.83_{\pm1.00}$ |
| Graphormer (Ying et al., 2021) | $80.51_{\pm0.53}$ |
| Cy2C-GCN (Choi et al., 2022) | $78.02_{\pm0.60}$ |
| GAWL (Nikolentzos et al., 2023) | $78.34_{\pm0.39}$ |
| LLM-GIN (Zhong et al., 2024) | $79.22_{\pm NA}$ |
| GMoE-GIN (Wang et al., 2023) | $76.90_{\pm0.90}$ |
| TopER (Tola et al., 2025) | $80.21_{\pm0.15}$ |
| TOPOFORMER* | $78.19_{\pm0.19}$ |

Against state-of-the-art supervised, contrastive, and fusion baselines, TOPOFORMER* achieves the *best ROC AUC* on *ToxCast, ClinTox*, and *BACE*, and is the *runner-up on Tox21* (Table 3). It remains competitive on HIV, trailing the leader by only a small margin. Aggregating across all seven benchmarks, TOPOFORMER attains the *second-lowest average deviation* from the column best (*AvD = 2.5*) and the *second-lowest average rank* (*AvR = 2.8*), confirming consistent top-tier performance

Table 5: **TOPOFORMER vs. PH:** Accuracy results for three topological models using degree centrality, Ollivier-Ricci and HKS filtrations. The PH-MLP model utilizes Betti vectors derived from regular sublevel filtrations combined with an MLP, while PH-TR applies transformers to the same vectors. The TOPOFORMER uses Betti sequences generated via the Topo-Scan on the same filtration function and applies transformers.

| Filtration | Model | BZR | COX2 | MUTAG | PROTEINS | IMDB-B | IMDB-M | REDDIT-B |
|---|---|---|---|---|---|---|---|---|
| **Degree** | PH-MLP | $82.71_{\pm6.51}$ | $76.44_{\pm5.39}$ | $84.06_{\pm4.65}$ | $68.37_{\pm3.97}$ | $65.70_{\pm4.03}$ | $45.07_{\pm2.59}$ | $89.50_{\pm2.87}$ |
| | PH-TR | $86.43_{\pm4.33}$ | $78.15_{\pm5.19}$ | $86.11_{\pm5.23}$ | $\mathbf{77.54_{\pm2.64}}$ | $\mathbf{75.00_{\pm2.11}}$ | $\mathbf{50.67_{\pm3.57}}$ | $\mathbf{92.30_{\pm1.77}}$ |
| | TOPOFORMER | $\mathbf{91.10_{\pm5.14}}$ | $\mathbf{80.27_{\pm5.24}}$ | $\mathbf{92.54_{\pm5.12}}$ | $77.45_{\pm4.02}$ | $74.20_{\pm5.01}$ | $50.33_{\pm1.52}$ | $89.75_{\pm2.18}$ |
| **O.Ricci** | PH-MLP | $85.45_{\pm3.36}$ | $78.16_{\pm5.09}$ | $84.06_{\pm5.21}$ | $65.50_{\pm4.26}$ | $68.00_{\pm3.55}$ | $44.87_{\pm3.65}$ | $85.65_{\pm2.62}$ |
| | PH-TR | $88.62_{\pm5.40}$ | $78.16_{\pm5.73}$ | $87.61_{\pm5.70}$ | $77.27_{\pm5.08}$ | $72.20_{\pm6.24}$ | $48.00_{\pm4.33}$ | $90.65_{\pm1.08}$ |
| | TOPOFORMER | $\mathbf{90.38_{\pm5.50}}$ | $\mathbf{80.72_{\pm6.44}}$ | $\mathbf{92.54_{\pm4.47}}$ | $\mathbf{77.90_{\pm3.17}}$ | $\mathbf{74.70_{\pm4.95}}$ | $\mathbf{51.53_{\pm3.49}}$ | $\mathbf{91.90_{\pm2.73}}$ |
| **HKS** | PH-MLP | $84.96_{\pm4.42}$ | $78.19_{\pm4.34}$ | $84.09_{\pm5.72}$ | $70.80_{\pm4.70}$ | $71.10_{\pm5.28}$ | $47.93_{\pm3.20}$ | $88.10_{\pm1.67}$ |
| | PH-TR | $89.60_{\pm5.84}$ | $79.89_{\pm4.66}$ | $94.12_{\pm5.42}$ | $77.18_{\pm3.15}$ | $76.80_{\pm3.97}$ | $53.60_{\pm3.31}$ | $87.25_{\pm1.95}$ |
| | TOPOFORMER | $\mathbf{90.62_{\pm4.91}}$ | $\mathbf{83.95_{\pm2.99}}$ | $\mathbf{95.32_{\pm5.58}}$ | $77.35_{\pm2.86}$ | $\mathbf{77.90_{\pm5.72}}$ | $\mathbf{54.07_{\pm2.54}}$ | $\mathbf{90.05_{\pm2.41}}$ |

alongside recent SOTA models such as KANO, MV-Mol, and MolFuse. We also benchmarked against hybrid classical (HC) models (Appendix A.3), where TOPOFORMER achieves the best result on four out of seven datasets and highly competitive results on others (Table 8). These findings highlight that transforming topology into compact, attention-ready tokens yields a robust and adaptable molecular predictor.

We further report *Hybrid Classical* baselines combining fingerprints, SMILES, and graph features with standard learners in Table 8. See Appendix A.3 for details of these models.

## 4.3 ABLATION STUDIES

We conduct **four ablation studies, as follows.**

**TOPOFORMER vs. PH** (Table 5): We compare TOPOFORMER with two persistent homology models using the same filtration functions and thresholds. *PH-MLP* uses sublevel filtrations with Betti vectorization followed by an MLP, while *PH-TR* replaces the MLP with a Transformer, treating Betti vectors as sequences. TOPOFORMER instead uses our proposed *Topo-Scan* to directly extract topological sequences. PH-TR outperforms PH-MLP, showing that sequential encodings preserve richer information than static features. TOPOFORMER further improves on PH-TR, indicating that Topo-Scan captures more expressive structure than standard PH filtrations.

**Effect of molecular fingerprints** (Table 7): We evaluate TOPOFORMER and Extended-Connectivity Fingerprints (ECFPs) both independently and in combination, including integration with PubChem descriptors. While topological and fingerprint models perform moderately on their own, their combination consistently outperforms individual baselines, suggesting that topological features complement domain-specific descriptors.

**Sensitivity to width parameter** (Table 6): We analyze how the sliding window size influences the performance of Topo-Scan. See Appendix C.5 for further details.

Table 6: **Width Parameter.** Performance comparison for different window width parameters across datasets.

| | | BZR | COX2 | MUTAG | PROTEINS | IMDB-B | IMDB-M | REDDIT-B |
|---|---|---|---|---|---|---|---|---|
| | $m=2$ | $\mathbf{89.89_{\pm3.74}}$ | $78.36_{\pm4.93}$ | $92.02_{\pm7.24}$ | $\mathbf{77.28_{\pm5.93}}$ | $74.20_{\pm3.36}$ | $\mathbf{51.53_{\pm3.34}}$ | $86.60_{\pm2.97}$ |
| Degree Centrality | $m=3$ | $88.64_{\pm5.30}$ | $\mathbf{78.38_{\pm5.04}}$ | $90.41_{\pm5.53}$ | $76.92_{\pm3.62}$ | $73.20_{\pm3.39}$ | $51.13_{\pm3.08}$ | $\mathbf{86.90_{\pm2.31}}$ |
| | $m=4$ | $88.86_{\pm4.33}$ | $78.16_{\pm6.07}$ | $\mathbf{92.57_{\pm5.63}}$ | $76.91_{\pm3.28}$ | $74.10_{\pm3.93}$ | $49.67_{\pm5.36}$ | $85.85_{\pm2.85}$ |
| | $m=2$ | $\mathbf{90.60_{\pm3.69}}$ | $\mathbf{78.60_{\pm4.79}}$ | $89.91_{\pm3.86}$ | $77.26_{\pm4.29}$ | $\mathbf{79.10_{\pm3.78}}$ | $\mathbf{54.53_{\pm3.52}}$ | $\mathbf{91.40_{\pm1.24}}$ |
| O. Ricci | $m=3$ | $89.14_{\pm4.70}$ | $78.15_{\pm5.73}$ | $\mathbf{91.02_{\pm6.45}}$ | $\mathbf{77.72_{\pm3.36}}$ | $78.80_{\pm3.79}$ | $53.73_{\pm4.06}$ | $89.95_{\pm2.24}$ |
| | $m=4$ | $88.39_{\pm6.44}$ | $78.17_{\pm5.05}$ | $89.85_{\pm6.40}$ | $77.35_{\pm4.05}$ | $78.10_{\pm3.14}$ | $53.87_{\pm5.27}$ | $89.65_{\pm2.37}$ |
| | $m=2$ | $90.62_{\pm4.91}$ | $83.95_{\pm2.99}$ | $\mathbf{95.32_{\pm5.58}}$ | $77.35_{\pm2.86}$ | $\mathbf{77.90_{\pm5.72}}$ | $\mathbf{54.07_{\pm2.54}}$ | $\mathbf{90.05_{\pm2.41}}$ |
| HKS | $m=3$ | $\mathbf{91.09_{\pm4.28}}$ | $\mathbf{85.01_{\pm4.84}}$ | $95.23_{\pm3.89}$ | $\mathbf{78.17_{\pm4.54}}$ | $76.90_{\pm3.48}$ | $53.60_{\pm3.30}$ | $88.80_{\pm1.86}$ |
| | $m=4$ | $90.63_{\pm4.09}$ | $83.75_{\pm5.09}$ | $95.12_{\pm6.05}$ | $78.07_{\pm2.84}$ | $77.00_{\pm4.62}$ | $53.40_{\pm3.51}$ | $89.00_{\pm1.80}$ |

**Single vs. multiple filtration functions** (Table 13): We test several node based and edge based functions to study how filtration choice affects performance. We observe that single-filtration

TopoFormer (for example, using only HKS or only Ollivier–Ricci) already achieves strong results, while combining filtrations yields modest but consistent improvements on some datasets. This indicates that multiple filtrations are a flexible way to incorporate complementary structural signals rather than a requirement for good performance.

**Discussion.** TOPOFORMER delivers consistently strong performance across a broad range of graph classification benchmarks, outperforming state-of-the-art baselines and achieving the best overall accuracy on most datasets. These results demonstrate the model's ability to extract essential structural information through topological patterns while producing *fixed-size sequential representations*. Such representations are particularly well-suited for Graph Foundation Models, which require consistent and transferable embeddings across graphs of varying sizes and domains. Table 2 further reveals that among the six topological baselines (PersLay, DMP, FC-V, EMP, MP-HSM, TopoGCL), TOPOFORMER achieves the best performance, despite being architecturally simpler and more computationally lightweight. This supports our design philosophy that robust topological summaries, when properly structured, can outperform more complex pipelines.

Crucially, TOPOFORMER departs from the standard GNN paradigm of first learning node embeddings followed by global pooling. While effective, this node-centric strategy treats graphs as unstructured point clouds in latent space, requiring repeated updates as embeddings evolve, often at the cost of coherence and efficiency (Mesquita et al., 2020; Liu et al., 2023). In contrast, topological models treat graphs as structured wholes and directly encode global patterns. By bypassing intermediate node embeddings, TOPOFORMER provides a streamlined and principled approach for learning stable, transferable graph-level representations.

**Limitations and future work.** Our focus in this work is on a streamlined, graph-level instantiation of TOPOFORMER, which also suggests several natural extensions. We restrict attention to low-dimensional homology ($H_0, H_1$) on a fixed clique complex with a small set of standard filtrations (degree, curvature, HKS); incorporating richer per-slice invariants or learnable filtrations could further boost expressivity while keeping the same Topo-Scan + Transformer backbone. Likewise, we concentrate on widely used graph-classification and molecular benchmarks, leaving node-/edge-level tasks and more heterogeneous settings (e.g., temporal or citation graphs) to future work. Finally, Topo-Scan is designed as a lightweight, scan-style summary that complements rather than replaces full persistent homology, and we see developing additional theory and applications for such summaries as an interesting direction for the TDA community.

## 5 CONCLUSION

Fixed-size, transferable representations remain a central challenge in graph learning. We introduce TOPOFORMER, a scalable framework that encodes multi-scale topological structure into attention-ready sequences. By replacing full persistence diagrams with lightweight, slice-wise invariants via Topo-Scan, our method integrates seamlessly with transformer architectures while offering theoretical stability guarantees. TOPOFORMER achieves state-of-the-art results across graph classification and molecular property prediction tasks, with predictable compute and strong generalization. Looking ahead, we aim to extend this framework toward graph foundation models by combining topological and spectral signals through large-scale self-supervised pretraining, and by adapting to dynamic and heterogeneous graphs via learnable filtrations.

ACKNOWLEDGMENTS

This work was partially supported by Canadian NSERC Discovery Grant RGPIN-2020-05665: Data Science on Blockchains, National Science Foundation under grants DMS-2220613, and DMS-2229417. The authors acknowledge the Texas Advanced Computing Center (TACC) at UT Austin for providing computational resources that have contributed to the research results reported within this paper.

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

# Appendix

## A TOPOFORMER*: TOPOFORMER FOR MPP

### A.1 MOLECULAR FINGERPRINTS

Molecular fingerprints are widely used in computational chemistry and machine learning to represent molecular structures as fixed-length numerical vectors (Cereto-Massagué et al., 2015). They encode features such as atomic connectivity and substructural patterns, enabling efficient similarity search and predictive modeling. Popular methods include ECFP (Extended Connectivity Fingerprints) and PubChemFP, both extensively applied in drug discovery, virtual screening, and bioinformatics (Yang et al., 2022).

**ECFP Fingerprints.** Extended Connectivity Fingerprints (ECFP) capture structural features by iteratively hashing local atomic environments up to a specified radius (Rogers & Hahn, 2010). Unlike traditional hashed fingerprints, ECFP preserves substructural detail, making it effective for similarity search, QSAR modeling, and property prediction. It is invariant to atom ordering while retaining connectivity, enabling fine-grained molecular feature analysis. For a recent overview of ECFP fingerprints and their role in modern biochemical ML pipelines, see (Wigh et al., 2022).

### A.2 TOPOFORMER* MODEL

For Molecular Property Prediction Task, we employ a hybrid model, TOPOFORMER*, combining ECFP Fingerprints and our TOPOFORMER Model. This hybrid model shows the versatility of our TOPOFORMER model on its effective integration with complementary information (Table 7). We give the flowchart of our hybrid model in Figure 5. In our hybrid model, we used the same experimental setup for the TOPOFORMER component. For the MLP component, we employed a two-layer MLP with a hidden dimension of 200, ensuring that its output dimension matches the output dimension of the TOPOFORMER model. The model was optimized using the Adam optimizer with a learning rate of 0.01 and a weight decay of 1e-4. Both the MLP and TOPOFORMER components were trained in an end-to-end manner, allowing the model to leverage both topological signatures and complementary graph information, ultimately leading to improved performance.

Table 7: Performance comparison of standalone models (TOPOFORMER and FP-MLP) and the hybrid model (TOPOFORMER*) in random splitting (8:1:1).

|         | PH-TR | TOPOFORMER | FP-MLP | PH+ECFP+TR | TOPOFORMER* |
|---------|-------|------------|--------|------------|-------------|
| BACE    | $72.41_{\pm3.15}$ | $83.29_{\pm2.14}$ | $90.29_{\pm2.67}$ | $90.60_{\pm2.99}$ | $\mathbf{91.60_{\pm1.73}}$ |
| HIV     | $69.29_{\pm1.65}$ | $75.81_{\pm0.23}$ | $83.26_{\pm1.01}$ | $83.97_{\pm1.51}$ | $\mathbf{85.10_{\pm0.49}}$ |
| BBBP    | $83.37_{\pm3.90}$ | $94.54_{\pm1.01}$ | $89.68_{\pm3.46}$ | $93.47_{\pm2.53}$ | $\mathbf{95.90_{\pm0.28}}$ |
| ClinTox | $75.89_{\pm6.60}$ | $83.42_{\pm2.33}$ | $76.34_{\pm6.54}$ | $82.04_{\pm7.12}$ | $\mathbf{86.20_{\pm3.83}}$ |
| SIDER   | $62.91_{\pm3.49}$ | $62.10_{\pm1.44}$ | $65.30_{\pm0.99}$ | $\mathbf{66.99_{\pm1.85}}$ | $66.80_{\pm0.29}$ |
| Tox21   | $68.24_{\pm1.60}$ | $80.87_{\pm0.19}$ | $77.89_{\pm1.54}$ | $79.03_{\pm1.21}$ | $\mathbf{81.50_{\pm1.85}}$ |
| ToxCast | $64.74_{\pm2.29}$ | $73.37_{\pm1.42}$ | $74.69_{\pm1.33}$ | $75.73_{\pm1.59}$ | $\mathbf{78.40_{\pm1.57}}$ |

### A.3 HYBRID CLASSICAL MPP BASELINES

We refer to the classical models combined with modern ML models as *Hybrid Classical (HC) Models*. The first family of HC baseline models consists of *Fingerprinting models* (Jiang et al., 2021), which use vectorized molecular fingerprints as input to traditional machine learning models, including SVM, XGB, RF, and MLP. The input fingerprints are a concatenation of 881-dimensional PubChem fingerprints (PubChemFP), 307-dimensional substructure fingerprints (SubFP), and 206-dimensional MOE 1-D and 2-D descriptors (Yap, 2011). The second family of baseline models comprises *SMILES models*, which treat SMILES strings as sequential input to 1D CNN (Kimber et al., 2021), a 3-layer bidirectional GRU (Cho et al., 2014), and a pre-trained SMILES transformer (TRSF) (Honda et al., 2019). The third family is *GNN models* which use 2D graph-based representations of compounds, where atom and bond features are encoded using one-hot schemes and fed into GCN, MPNN, GAT, and AFP models (Xiong et al., 2019). Another baseline is the SPN model, using SphereNet (Liu et al., 2022b), which employs 3D graphs of compounds as input.

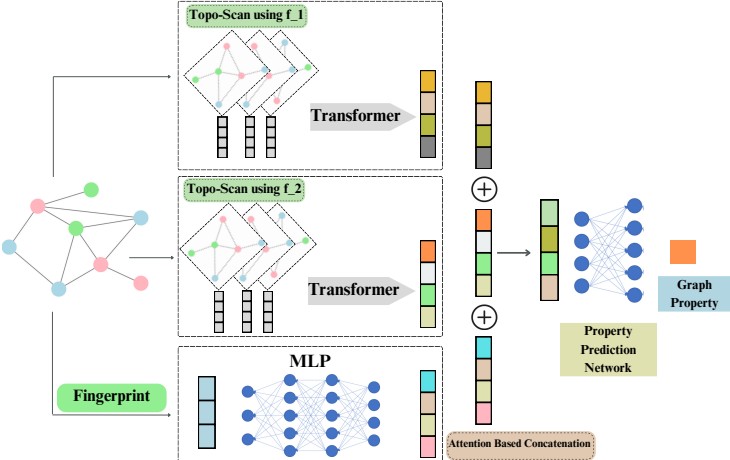

Figure 5: TOPOFORMER*: To successfully integrate complementary graph information, such as ECFP, with our TOPOFORMER model, we employ a MLP. The MLP output is combined with the TOPOFORMER model using an attention mechanism, and the combined representation is then passed through a graph prediction network to perform the final prediction task.

Table 8: **Hybrid Classical MPP Models.** The ROC AUC results of ML models for molecular property prediction tasks with random splitting (8:1:1). The baseline results are reported from (Xia et al., 2023). The best and the second best performances are given in bold, and underlined, respectively.

| | **Fingerprinting Models** | | | | **SMILES Models** | | | **GNN Models** | | | | | **Ours** |
| **Dataset** | **SVM** | **XGB** | **RF** | **CNN** | **RNN** | **TRSF** | **MLP** | **GCN** | **MPNN** | **GAT** | **AFP** | **SPN** | **TOPOFORMER*** |
|---|---|---|---|---|---|---|---|---|---|---|---|---|---|
| BBBP | 91.3 | 92.6 | 92.3 | 89.7 | 76.0 | 69.3 | 89.7 | 91.8 | 91.5 | 87.2 | 90.2 | 90.5 | **96.6** |
| Tox21 | 82.0 | 83.7 | 83.1 | 81.2 | 73.7 | 76.8 | 79.9 | **84.6** | 82.1 | 84.5 | 82.7 | 82.5 | 81.5 |
| ToxCast | 72.5 | 78.5 | 77.8 | 73.5 | 67.8 | 78.0 | 78.1 | 76.7 | **78.8** | 77.2 | 76.8 | 77.2 | 78.4 |
| SIDER | 62.6 | 63.8 | 64.4 | 59.1 | 51.5 | 64.1 | 61.7 | 62.3 | 60.3 | 62.0 | 61.3 | 61.3 | **66.8** |
| ClinTox | 87.9 | 91.9 | 93.0 | 88.8 | 68.5 | **96.3** | 93.0 | 88.9 | 86.8 | 89.8 | 87.9 | 91.2 | 86.2 |
| BACE | 88.6 | 89.6 | 89.0 | 81.5 | 55.9 | 83.5 | 88.7 | 88.0 | 84.6 | 88.6 | 87.9 | 88.2 | **91.6** |
| HIV | 81.7 | 83.9 | 82.0 | 82.6 | 73.3 | 74.8 | 79.1 | 83.4 | 81.4 | 81.2 | 81.8 | 81.8 | **85.1** |

# B  PROOFS OF STABILITY THEOREMS

We work on the fixed clique complex $\widehat{G}$ of $G = (V, E)$. For a node function $h : V \to \mathbb{R}$, we use the upper–star extension $\widehat{h}(\sigma) = \max_{v \in \sigma} h(v)$ and the associated sublevel filtration on $\widehat{G}$. Throughout, $k \in \{0, 1\}$ is the homological dimension used in our tokens.

**Preliminaries.** For $a \le b$, define the interlevel (level-set) subcomplex $\widehat{G}^h_{[a,b]} := \{\sigma \in \widehat{G} : a \le \min_{v \in \sigma} h(v) \text{ and } \max_{v \in \sigma} h(v) \le b\}$. The associated pointwise finite-dimensional *interlevel persistence module* is the functor $M^h_k : (a, b) \mapsto H_k(\widehat{G}^h_{[a,b]})$. Given a shared grid $\alpha_0 < \cdots < \alpha_N$, window width $m$, and stride $s$, the Topo-Scan token at window $t$ is

$$\widehat{\beta}^h_k(t) \; = \; \dim M^h_k\big(\alpha_{ts}, \alpha_{ts+m}\big), \quad t = 0, \ldots, T-1, \quad T = \left\lfloor \tfrac{N-m}{s} \right\rfloor + 1.$$

We write $d_B(M^f_k, M^g_k)$ for the bottleneck distance between the interval decompositions (barcodes) of the interlevel modules $M^f_k$ and $M^g_k$.

**Two stability lemmas.**

**Lemma B.1** (Interlevel stability). *(Botnan & Lesnick, 2018, Thm 1.1 & 1.2) For $k \ge 0$, the interlevel modules of the upper–star filtrations induced by $f, g : V \to \mathbb{R}$ on the fixed clique complex $\widehat{G}$ satisfy*

$$d_B\big(M^f_k, M^g_k\big) \; \le \; \|f - g\|_\infty.$$

**Lemma B.2** (Lipschitzness of interval rank). *(Bauer & Lesnick, 2014; Bakke Bjerkevik, 2021) Let $M, N$ be interval-decomposable, p.f.d. modules with $d_B(M, N) \leq \delta$. For any interval $I = [a, b]$,*

$$\left| \dim M(a, b) - \dim N(a, b) \right| \leq \mathcal{B}_M(I, \delta) + \mathcal{B}_N(I, \delta),$$

*where $\mathcal{B}_M(I, \delta)$ counts bars in $\mathrm{Bar}(M)$ whose endpoints lie within $\delta$ of the boundary $\{a, b\}$ (and similarly for $N$).*

**Theorem 3.1.** *With the setup above, there exists $C = C(\widehat{G}, \{\alpha_i\}, m, s)$ such that*

$$\sum_{t=0}^{T-1} \left| \widehat{\beta}_k^f(t) - \widehat{\beta}_k^g(t) \right| \leq C \, d_B(M_k^f, M_k^g).$$

*Proof of Theorem 3.1.* Fix $t$ and write $I_t = [\alpha_{ts}, \alpha_{ts+m}]$. By Lemma B.2, there exists a finite constant $C_0(\widehat{G}, I_t)$ such that $|\dim M_k^f(I_t) - \dim M_k^g(I_t)| \leq C_0(\widehat{G}, I_t) \, d_B(M_k^f, M_k^g)$. Summing over $t$ gives

$$\sum_{t=0}^{T-1} \left| \widehat{\beta}_k^f(t) - \widehat{\beta}_k^g(t) \right| \leq \left( \sum_{t=0}^{T-1} C_0(\widehat{G}, I_t) \right) d_B(M_k^f, M_k^g) := C \, d_B(M_k^f, M_k^g).$$

On a fixed finite complex and fixed grid, the $C_0(\widehat{G}, I_t)$ are finite and can be uniformly bounded, yielding $C = T \, C_0$. □

**Corollary 3.2.** *For upper–star filtrations on a fixed complex, $d_B(M_k^f, M_k^g) \leq \|f - g\|_\infty$ (Lemma B.1), hence $\|\widehat{\beta}_k(G, f) - \widehat{\beta}_k(G, g)\|_1 \leq C \|f - g\|_\infty$.*

*Proof of Theorem 3.1.* By Theorem 3.1, we have $\|\widehat{\beta}_k(G, f) - \widehat{\beta}_k(G, g)\|_1 \leq C \, d_B(M_k^f, M_k^g)$.

By Lemma B.1 (interlevel/level-set stability on the fixed clique complex), $d_B(M_k^f, M_k^g) \leq \|f - g\|_\infty$. Combining the two inequalities yields the claim. □

**Connection to classical sublevel stability.** The inequality $d_B \leq \|f - g\|_\infty$ is classical for sublevel filtrations on a fixed space (Cohen-Steiner et al., 2007). Our Lemma B.1 is the level-set (interlevel) analogue on the fixed clique complex, following algebraic stability for zigzag/level-set modules (e.g., Botnan & Lesnick, 2018). We use this interlevel version to handle windowed intervals $[a, b]$ appearing in Topo-Scan.

**Shared thresholds.** The theorem assumes a shared grid $\{\alpha_i\}$. If thresholds are chosen separately (e.g., per-function quantiles), a monotone reparameterization of the filtration axis induces an additional term proportional to the grid displacement, which can be absorbed into $C$.

*Remark* B.3 (Relation to PH invariants and stable ranks). Our stability theorem focuses on the $\ell_1$ robustness of the discrete Topo-Scan sequences $\widehat{\beta}_k(G, h)$, but these sequences implicitly encode familiar PH objects. For a fixed filtration function $h$, the map $t \mapsto \widehat{\beta}_k^h(t)$ can be viewed as a sampled version of the rank invariant $(a, b) \mapsto \mathrm{rank} \, H_k\big((\widehat{G})_{[a,b]}^h\big)$ associated with the interlevel module $M_k^h$. In this sense, Topo-Scan produces a coarse, structured discretization of the same information that barcodes and stable vectorizations of persistence diagrams, such as persistence landscapes, silhouettes and persistence images (Chazal et al., 2014; Adams et al., 2017), summarize in continuous form. Similarly, Graph Filtration Learning (Hofer et al., 2020) can be seen as learning the filtration function $h$, while our work fixes $h$ and instead changes the representation from global barcodes to local interlevel sequences. A full expressivity comparison and formal information-loss bounds relative to complete barcodes are interesting directions for future work.

## C  MORE ON TOPOFORMER

### C.1  BASE MODEL: TRANSFORMER

Our TOPOFORMER model is designed for classification tasks using sequential inputs, harnessing transformers for efficient feature extraction. The architecture includes an embedding layer, a transformer encoder, and a fully connected (FC) classification head, with regularization techniques applied to mitigate overfitting.

Let $\mathbf{x} = (x_1, x_2, \ldots, x_T) \in \mathbb{R}^{N \times T \times D}$ represent the input sequence, where $N$ is number of graphs, $T$ is the sequence length, and $D$ is the dimensionality of each input token. The input sequence is first passed through an embedding layer $\mathbf{E} : \mathbb{R}^D \to \mathbb{R}^H$, where $H$ denotes the embedding dimension. In addition, a positional encoding matrix $\mathbf{P} \in \mathbb{R}^{1 \times T \times H}$ is added to the embeddings to encode the positional information of the sequence, resulting in a sequence of embedded vectors $\mathbf{e}_t = \mathbf{E}(x_t) + \mathbf{P}_t$ for $t = 1, 2, \ldots, T$, where $\mathbf{P}_t$ is the positional encoding for position $t$.

The sequence of embeddings is then passed through a multi-layer transformer encoder, where the encoder operates on the embedded sequence $\mathbf{E}(\mathbf{x}) + \mathbf{P} \in \mathbb{R}^{T \times B \times H}$, with $B$ representing the batch size. The transformer encoder generates a new sequence of output representations $\mathbf{z} = (z_1, z_2, \ldots, z_T) \in \mathbb{R}^{T \times B \times H}$. After processing through the encoder, the output sequence is permuted and reshaped to a flattened vector of size $B \times (T \cdot H)$, ensuring compatibility with subsequent fully connected layers.

The flattened representation $\mathbf{z}_{\text{flat}} \in \mathbb{R}^{B \times (T \cdot H)}$ is then passed through a batch normalization layer, $\mathbf{BN}(\mathbf{z}_{\text{flat}})$, which normalizes the activations across the batch to stabilize the training process. A dropout layer $\mathbf{D}(\cdot)$ is then applied to the normalized output to regularize the model and mitigate overfitting. The final classification output is obtained through a fully connected layer $\mathbf{FC} : \mathbb{R}^{B \times (T \cdot H)} \to \mathbb{R}^H$.

## C.2 DUAL TRANSFORMER WITH MULTI-LAYER PERCEPTRON CLASSIFIER

This model combines multiple sources of input data through a hybrid architecture that integrates two independent base models and a multi-layer perceptron (MLP). This model is designed to handle diverse input modalities by leveraging the strengths of both transformers and MLPs for feature extraction and classification.

Let $\mathbf{X}_1 \in \mathbb{R}^{N \times T_1 \times D_1}$, $\mathbf{X}_2 \in \mathbb{R}^{N \times T_2 \times D_2}$, and $\mathbf{X}_3 \in \mathbb{R}^{N \times L}$ represent the three distinct input graph encoding, where $T_i$ denotes the sequence length, $D_i$ the dimensionality of the inputs for each modality and $L$ the dimension of fingerprints. Each input is processed through its respective component: the first sequence $\mathbf{X}_1$ is passed through transformer $\mathcal{T}_1$, the second sequence $\mathbf{X}_2$ through transformer $\mathcal{T}_2$, and the third sequence $\mathbf{X}_3$ through an MLP $\mathcal{M}$.

The output of the first transformer $\mathcal{T}_1$, denoted $\mathbf{z}_1 \in \mathbb{R}^{T_1 \times B \times H}$, is obtained by passing $\mathbf{X}_1$ through the transformer encoder. Similarly, the output of the second transformer $\mathcal{T}_2$, denoted $\mathbf{z}_2 \in \mathbb{R}^{T_2 \times B \times H}$, is obtained by processing $\mathbf{X}_2$. Finally, the output of the MLP $\mathcal{M}$ is denoted $\mathbf{z}_3 \in \mathbb{R}^{B \times H}$.

The outputs $\mathbf{z}_1$, $\mathbf{z}_2$, and $\mathbf{z}_3$ are then combined through a learnable weighted sum. Specifically, the combined feature vector $\mathbf{z}_{\text{combined}}$ is computed as:

$$\mathbf{z}_{\text{combined}} = \alpha \cdot \mathbf{z}_1 + \beta \cdot \mathbf{z}_2 + (1 - \alpha - \beta) \cdot \mathbf{z}_3$$

where $\alpha$ and $\beta$ are learnable parameters that control the contribution of each modality to the final representation. This weighted combination allows the model to adaptively learn the most relevant contribution of each input sequence.

The combined feature vector $\mathbf{z}_{\text{combined}}$ is then passed through a batch normalization layer $\mathbf{BN}(\mathbf{z}_{\text{combined}})$ to normalize the activations, improving training stability. A final fully connected layer $\mathbf{FC}$ produces the classification output: $\hat{y} = \mathbf{FC}(\mathbf{z}_{\text{combined}})$ where $\hat{y} \in \mathbb{R}^C$ represents the predicted class probabilities, with $C$ being the number of possible output classes.

## C.3 RUNTIME ANALYSIS

To assess the computational efficiency of our method, we report the total runtime across two key stages: (i) topological signature extraction using Topo-Scan (via Degree Centrality and Ollivier-Ricci curvature), and (ii) model training using the transformer-based classifier. Table 9 presents a detailed breakdown of runtimes (in minutes) for five benchmark datasets.

As expected, Degree Centrality is extremely fast to compute and contributes negligible overhead. Ollivier-Ricci curvature, while more computationally intensive, remains tractable even for large graphs, as evidenced by reasonable runtimes on datasets such as REDDIT-5K and OGBG-MOLHIV. Transformer training times scale smoothly with dataset size and remain within practical limits.

Overall, our method maintains scalability while offering strong performance, demonstrating the feasibility of integrating topological signatures into deep graph models at scale.

**TopoScan vs. PH.** We report in Table 10 the runtime for Topo-Scan and standard PH on four benchmark datasets with degree centrality filtration (already computed), using the same backend (pyflagser) for both pipelines. For Topo-Scan, we invoke the unweighted flagser routine, since our method only requires Betti numbers on unweighted clique complexes. For PH, we use the weighted flagser routine, which constructs a full filtration and computes persistence diagrams.

Table 9: **Runtimes.** Total runtime (in minutes) per dataset. The second and third columns report the time to compute scalar filtration values and Topo-Scan vectorizations for degree centrality and O.Ricci curvature, respectively, and the final column shows the Transformer training time.

| Dataset | Degree C. | O. Ricci | Transformer |
|---------|-----------|----------|-------------|
| IMDB-B | 0.51 | 5.04 | 3.05 |
| IMDB-M | 0.49 | 7.62 | 4.57 |
| REDDIT-B | 5.30 | 23.70 | 6.10 |
| REDDIT-5K | 36.74 | 109.98 | 15.65 |
| OGBG-MOLHIV | 14.70 | 339.06 | 21.67 |

The clustering coefficient column serves as a proxy for graph density and hence clique complexity. On highly clustered graphs such as IMDB-B and IMDB-M (coefficients $\approx 0.95$–$0.97$), PH is roughly $13$–$14\times$ slower than Topo-Scan, reflecting the combinatorial explosion of cliques and the cost of global boundary-matrix reductions, whereas on the sparser REDDIT datasets the gap is smaller but still consistent (about $2\times$ on REDDIT-B and $1.5\times$ on REDDIT-5K). These results empirically confirm that Topo-Scan achieves multi-fold runtime savings over standard PH pipelines on dense graphs while remaining uniformly more efficient across all tested datasets.

Table 10: **Runtime for PH vs. Topo-Scan.** Runtime (in seconds) per dataset for computing topological features using the same backend (pyflagser).

| Dataset | Clus. Coeff. | Topo-Scan | PH |
|---------|--------------|-----------|-----|
| IMDB-B | 0.947 | 9.67 | 135.48 |
| IMDB-M | 0.969 | 17.81 | 234.30 |
| REDDIT-B | 0.048 | 12.29 | 24.25 |
| REDDIT-5K | 0.027 | 29.51 | 43.92 |

**Comparison with other methods.** We also compare the runtimes of two PH-based baselines, PersLay (with degree centrality input) and TopoGCL (using only the topology derived component), against Topo-Scan on IMDB-B and REDDIT-B (Table 11). All times are reported in seconds. For Topo-Scan, we include both the scalar filtration computation and Topo-Scan feature extraction.

Table 11: Runtimes (in seconds) for PH-based baselines (PersLay, TopoGCL) and Topo-Scan on IMDB-B and REDDIT-B.

| Method | IMDB-B | REDDIT-B | Notes |
|--------|--------|----------|-------|
| PersLay | 97.02 | 454.78 | PH-based layer on degree input |
| TopoGCL | 435.49 | 4010.08 | Only topological component |
| Topo-Scan | 15.16 | 290.00 | Filtration + Topo-Scan |

On IMDB-B, Topo-Scan is about 6 times faster than PersLay and around 29 times faster than the topological part of TopoGCL; on REDDIT-B, it remains faster than PersLay and roughly 14 times faster than TopoGCL. These results further support the practical efficiency of Topo-Scan compared with PH-based pipelines.

## C.4 Comparison with Pooling Methods

Table 12 compares TOPOFORMER with six representative graph pooling methods designed to adapt GNNs to graph-level tasks. DiffPool (Ying et al., 2018) learns a soft assignment matrix that hierarchically clusters nodes in a differentiable, end-to-end manner. Top-KPooling with Graph U-Nets (Top-K) (Gao & Ji, 2019) ranks nodes using a learnable projection score and retains the top-$k$ fraction to coarsen the graph. EigenPool (EigenGCN) (Ma et al., 2019) projects node features onto the leading eigenvectors of the graph Laplacian to preserve global spectral properties. SAGPool (Lee et al., 2019) computes attention scores through a GNN layer, pruning low-importance nodes and re-wiring the remaining graph. MinCutPool (Bianchi et al., 2020) casts pooling as a relaxed spectral clustering problem by optimizing a minimum-cut objective to form node clusters. HaarPool (Wang et al., 2020) applies a Haar wavelet transform to graph signals and performs pooling by selecting key wavelet coefficients. Our model TOPOFORMER takes a different approach by integrating multiscale topological filtrations with a transformer-based attention mechanism, enabling the pooling of substructures across scales and yielding robust higher-order graph representations. As shown in Table 12, TOPOFORMER consistently outperforms all baselines, achieving the best accuracy on six out of seven datasets and ranking second on the remaining one.

Table 12: **Comparison with Pooling Methods.** Accuracy results of six baseline pooling methods and TOPOFORMER on seven graph classification benchmark datasets.

| Model | BZR | COX2 | MUTAG | PROTEINS | IMDB-B | IMDB-M | REDDIT-B |
|---|---|---|---|---|---|---|---|
| Top-K | $79.40_{\pm1.20}$ | $80.30_{\pm4.21}$ | $67.61_{\pm3.36}$ | $69.60_{\pm3.50}$ | $73.17_{\pm4.84}$ | $48.80_{\pm3.19}$ | $79.40_{\pm7.40}$ |
| MinCutPool | $82.64_{\pm5.05}$ | $80.07_{\pm3.85}$ | $79.17_{\pm1.64}$ | $\underline{76.52}_{\pm2.58}$ | $70.77_{\pm4.89}$ | $49.00_{\pm2.83}$ | $\underline{87.20}_{\pm5.00}$ |
| DiffPool | $83.93_{\pm4.41}$ | $79.66_{\pm2.64}$ | $79.22_{\pm1.02}$ | $73.63_{\pm3.60}$ | $68.60_{\pm3.10}$ | $45.70_{\pm3.40}$ | $79.00_{\pm1.10}$ |
| EigenGCN | $83.05_{\pm6.00}$ | $80.16_{\pm5.80}$ | $79.50_{\pm0.66}$ | $74.10_{\pm3.10}$ | $70.40_{\pm3.30}$ | $47.20_{\pm3.00}$ | N/A |
| SAGPool | $82.95_{\pm4.91}$ | $79.45_{\pm2.98}$ | $76.78_{\pm2.12}$ | $71.86_{\pm0.97}$ | $\underline{74.87}_{\pm4.09}$ | $49.33_{\pm4.90}$ | $84.70_{\pm4.40}$ |
| HaarPool | $\underline{83.95}_{\pm5.68}$ | $\underline{82.61}_{\pm2.69}$ | $\underline{90.00}_{\pm3.60}$ | $73.23_{\pm2.51}$ | $73.29_{\pm3.40}$ | $\underline{49.98}_{\pm5.70}$ | N/A |
| TOPOFORMER | $\mathbf{92.36}_{\pm4.11}$ | $\mathbf{83.93}_{\pm4.03}$ | $\mathbf{94.68}_{\pm4.30}$ | $\mathbf{77.64}_{\pm3.64}$ | $\mathbf{78.90}_{\pm3.31}$ | $\mathbf{55.40}_{\pm4.78}$ | $\mathbf{91.50}_{\pm1.89}$ |

## C.5 TOPO-SCAN HYPERPARAMETERS

In the Topo-Scan algorithm, two key hyperparameters play a crucial role: the width parameter, which controls the thickness of slices, and the filtration function, which defines the hierarchical importance of nodes or edges. To determine the optimal hyperparameter settings, we conducted extensive experiments to validate their impact on model performance.

**Width Parameter Selection.** To determine the optimal width parameter $m$, we conducted experiments using degree centrality and Ollivier-Ricci curvature as filtration functions for the Topo-Scanner on graph classification datasets. We evaluated $m = 2, 3$, and $4$, extracting the corresponding Topo-Scanner feature vectors and using them as inputs to a transformer model. The results presented in Table 6 indicate that, for most of the datasets, the Topo-Scanner features achieve the best performance when $m = 2$ for both filtration functions. Based on this experimental analysis, we select $m = 2$ as the optimal parameter for our model.

**Multiple Filtrations.** Different filtration functions impose distinct hierarchical orderings on nodes (or edges), enabling our model to capture diverse topological patterns in the induced sequences. This allows the *Topo-Scan* process to effectively integrate domain-specific information. To fully leverage multiple filtrations, TOPOFORMER applies separate transformers for each filtration function and combines their outputs using a learnable attention mechanism. This mechanism dynamically assigns higher weights to the most relevant topological signatures, ensuring optimal feature selection and enhanced performance. As shown in Table 13, TOPOFORMER employing multiple functions consistently outperforms models using a single filtration function, demonstrating the advantages of multiple filtrations.

This approach enhances model robustness and stability by incorporating diverse topological perspectives.

Table 13: **Filtration Functions.** Performance comparison of single filtration and multiple filtrations with TOPOFORMER across different datasets. The best values in each column are highlighted in bold.

| Filtrations | MUTAG | PROTEINS | BZR | COX2 | IMDB-B | IMDB-M | REDDIT-B |
|---|---|---|---|---|---|---|---|
| Degree only | $92.02_{\pm7.24}$ | $77.28_{\pm5.93}$ | $89.89_{\pm3.74}$ | $78.36_{\pm4.93}$ | $74.20_{\pm3.36}$ | $51.53_{\pm3.34}$ | $86.60_{\pm2.97}$ |
| O. Ricci only | $89.91_{\pm3.86}$ | $77.26_{\pm4.29}$ | $90.60_{\pm3.69}$ | $78.60_{\pm4.79}$ | $\mathbf{79.10}_{\pm3.78}$ | $\mathbf{54.53}_{\pm3.52}$ | $\mathbf{91.40}_{\pm1.24}$ |
| HKS Only | $\mathbf{95.32}_{\pm5.58}$ | $\mathbf{77.35}_{\pm2.86}$ | $90.62_{\pm4.91}$ | $\mathbf{83.95}_{\pm2.99}$ | $76.90_{\pm5.72}$ | $54.07_{\pm2.54}$ | $90.05_{\pm2.41}$ |
| Deg.+O.Ricci | $93.01_{\pm5.29}$ | $\mathbf{78.35}_{\pm4.22}$ | $\mathbf{91.12}_{\pm4.68}$ | $81.80_{\pm5.40}$ | $78.80_{\pm3.65}$ | $53.87_{\pm3.52}$ | $90.65_{\pm2.12}$ |
| HKS+O.Ricci | $94.68_{\pm4.30}$ | $77.64_{\pm3.64}$ | $92.36_{\pm4.11}$ | $\mathbf{83.93}_{\pm4.03}$ | $78.90_{\pm3.31}$ | $\mathbf{55.40}_{\pm4.78}$ | $\mathbf{91.50}_{\pm1.89}$ |
| HKS+Degree | $\mathbf{95.26}_{\pm3.88}$ | $78.08_{\pm2.34}$ | $91.09_{\pm5.53}$ | $83.71_{\pm4.38}$ | $77.30_{\pm2.41}$ | $52.60_{\pm2.25}$ | $89.90_{\pm2.35}$ |

## C.6 TOPOFORMER VS. PH WITH DIFFERENT VECTORIZATIONS

TOPOFORMER consistently outperforms Persistent Homology methods in both accuracy and computational efficiency. As shown in Table 14, we compare against the best PH results reported in (Cai & Wang, 2020), which evaluates 16 combinations of four filtration functions (degree, O.Ricci, Fiedler, closeness centrality) and four vectorization techniques (Sliced Wasserstein, Pervec, Filvec, SW-p) per dataset. TOPOFORMER *achieves higher accuracy on all six benchmarks*.

Table 14: Accuracy results for TOPOFORMER (HKS) vs. Persistent Homology in graph classification tasks. In PH row, we report the best performance of 16 combinations with four filtration functions combined with four vectorizations.

| | BZR | COX2 | PROTEINS | IMDB-B | IMDB-M |
|---|---|---|---|---|---|
| PH (Best of 16 comb) | $88.4_{\pm0.6}$ | $82.0_{\pm0.6}$ | $74.0_{\pm0.4}$ | $69.5_{\pm0.5}$ | $46.5_{\pm0.3}$ |
| TOPOFORMER | $90.6_{\pm4.9}$ | $82.0_{\pm4.6}$ | $77.4_{\pm2.9}$ | $77.9_{\pm3.4}$ | $54.1_{\pm2.5}$ |

Table 15: **TOPOFORMER vs. PH Performance Comparison:** Accuracy of three topological models under seven filtrations: Degree, Ollivier-Ricci, HKS, Betweenness centrality, Closeness centrality, Eigenvector centrality, and Forman-Ricci curvature. The last column reports the average accuracy improvements of our models PH-TR and TOPOFORMER over the classical TDA pipeline PH-MLP for the same filtration function.

| Filtration | Model | BZR | COX2 | MUTAG | PROTEINS | IMDB-B | IMDB-M | REDDIT-B | Av.Imp. |
|---|---|---|---|---|---|---|---|---|---|
| **Degree** | PH-MLP | $82.71_{\pm6.51}$ | $76.44_{\pm5.39}$ | $84.06_{\pm4.65}$ | $68.37_{\pm3.97}$ | $65.70_{\pm4.03}$ | $45.07_{\pm2.59}$ | $89.50_{\pm2.87}$ | – |
| | PH-TR | $86.43_{\pm4.33}$ | $78.15_{\pm5.19}$ | $86.11_{\pm5.23}$ | $77.54_{\pm2.64}$ | $75.00_{\pm2.11}$ | $50.67_{\pm3.57}$ | $92.30_{\pm1.77}$ | 4.91 |
| | TOPOFORMER | $91.10_{\pm5.14}$ | $80.27_{\pm5.24}$ | $92.54_{\pm5.12}$ | $77.45_{\pm4.02}$ | $74.20_{\pm5.01}$ | $50.33_{\pm1.52}$ | $89.75_{\pm2.18}$ | 6.26 |
| **O.Ricci** | PH-MLP | $85.45_{\pm3.36}$ | $78.16_{\pm5.09}$ | $84.06_{\pm5.21}$ | $65.50_{\pm4.26}$ | $68.00_{\pm3.55}$ | $44.87_{\pm3.65}$ | $85.65_{\pm2.62}$ | – |
| | PH-TR | $88.62_{\pm5.40}$ | $78.16_{\pm5.73}$ | $87.61_{\pm5.70}$ | $77.27_{\pm5.08}$ | $72.20_{\pm6.24}$ | $48.00_{\pm4.33}$ | $90.65_{\pm1.08}$ | 4.40 |
| | TOPOFORMER | $90.38_{\pm5.50}$ | $80.72_{\pm6.44}$ | $92.54_{\pm4.47}$ | $77.90_{\pm3.17}$ | $74.70_{\pm4.95}$ | $51.53_{\pm3.49}$ | $91.90_{\pm2.73}$ | 6.85 |
| **HKS** | PH-MLP | $84.96_{\pm4.42}$ | $78.19_{\pm4.34}$ | $84.09_{\pm5.72}$ | $70.80_{\pm4.70}$ | $71.10_{\pm5.28}$ | $47.93_{\pm3.20}$ | $88.10_{\pm1.67}$ | – |
| | PH-TR | $89.60_{\pm5.84}$ | $79.89_{\pm4.66}$ | $94.12_{\pm5.42}$ | $77.18_{\pm3.15}$ | $76.80_{\pm3.97}$ | $53.60_{\pm3.31}$ | $87.25_{\pm1.95}$ | 4.75 |
| | TOPOFORMER | $90.62_{\pm4.91}$ | $83.95_{\pm2.99}$ | $95.32_{\pm5.58}$ | $77.35_{\pm2.86}$ | $77.90_{\pm5.72}$ | $54.07_{\pm2.54}$ | $90.05_{\pm2.41}$ | 6.30 |
| **Betweenness** | PH-MLP | $84.95_{\pm4.19}$ | $80.99_{\pm6.35}$ | $89.94_{\pm7.93}$ | $71.61_{\pm1.85}$ | $68.10_{\pm2.55}$ | $43.80_{\pm1.74}$ | $79.10_{\pm2.88}$ | – |
| | PH-TR | $85.43_{\pm4.13}$ | $81.60_{\pm6.00}$ | $90.52_{\pm5.62}$ | $74.13_{\pm2.95}$ | $69.90_{\pm3.48}$ | $45.40_{\pm1.79}$ | $84.05_{\pm2.33}$ | 1.79 |
| | TOPOFORMER | $87.41_{\pm4.07}$ | $80.74_{\pm6.15}$ | $90.99_{\pm6.61}$ | $76.73_{\pm2.67}$ | $73.90_{\pm3.73}$ | $51.47_{\pm2.96}$ | $86.55_{\pm2.30}$ | 4.19 |
| **Closeness** | PH-MLP | $84.21_{\pm2.19}$ | $79.07_{\pm6.56}$ | $88.94_{\pm7.20}$ | $74.39_{\pm3.19}$ | $65.30_{\pm4.61}$ | $47.47_{\pm3.93}$ | $66.85_{\pm2.98}$ | – |
| | PH-TR | $87.43_{\pm4.83}$ | $79.65_{\pm4.98}$ | $89.94_{\pm6.25}$ | $75.93_{\pm3.32}$ | $69.70_{\pm4.60}$ | $50.47_{\pm3.72}$ | $77.20_{\pm3.31}$ | 3.44 |
| | TOPOFORMER | $85.13_{\pm8.36}$ | $81.17_{\pm5.28}$ | $90.88_{\pm6.65}$ | $77.64_{\pm4.36}$ | $73.20_{\pm2.20}$ | $51.07_{\pm3.02}$ | $86.40_{\pm2.22}$ | 5.61 |
| **Eigenvector** | PH-MLP | $83.97_{\pm3.46}$ | $80.56_{\pm6.04}$ | $89.39_{\pm6.64}$ | $67.20_{\pm5.87}$ | $66.70_{\pm2.61}$ | $47.40_{\pm3.00}$ | $79.40_{\pm3.21}$ | – |
| | PH-TR | $87.41_{\pm4.21}$ | $79.88_{\pm6.04}$ | $91.57_{\pm5.70}$ | $70.53_{\pm4.60}$ | $72.40_{\pm4.48}$ | $50.13_{\pm3.44}$ | $89.25_{\pm1.40}$ | 3.79 |
| | TOPOFORMER | $90.59_{\pm5.63}$ | $82.87_{\pm3.35}$ | $90.99_{\pm6.61}$ | $77.35_{\pm2.78}$ | $76.10_{\pm3.63}$ | $51.00_{\pm2.14}$ | $91.85_{\pm1.43}$ | 6.59 |
| **F. Ricci** | PH-MLP | $82.46_{\pm3.94}$ | $80.13_{\pm5.86}$ | $87.81_{\pm6.21}$ | $73.95_{\pm4.12}$ | $66.60_{\pm3.75}$ | $45.53_{\pm3.36}$ | $73.70_{\pm3.78}$ | – |
| | PH-TR | $86.18_{\pm6.06}$ | $81.97_{\pm6.62}$ | $91.99_{\pm4.63}$ | $76.09_{\pm4.04}$ | $70.80_{\pm4.47}$ | $50.67_{\pm2.59}$ | $77.20_{\pm2.21}$ | 3.53 |
| | TOPOFORMER | $88.41_{\pm6.04}$ | $81.02_{\pm6.46}$ | $92.08_{\pm5.67}$ | $77.81_{\pm3.80}$ | $79.40_{\pm3.69}$ | $54.47_{\pm3.34}$ | $88.95_{\pm1.94}$ | 7.42 |

## C.7 TOPOFORMER VS PH PERFORMANCE

Table 15 extends our ablation (Table 5) from three to seven filtration functions and compares three topological pipelines under the same filtration function: the classical PH-MLP baseline (sublevel PH + Betti vector + MLP), our PH-TR variant (same Betti vectors but processed as sequences by a Transformer), and TOPOFORMER (Topo-Scan sequences with sliding-window interlevel filtrations). Across all seven filtrations, replacing the MLP with a Transformer already yields consistent gains: PH-TR improves over PH-MLP by roughly 2–4 accuracy points on average (see the "Av.Imp." column), confirming that treating Betti curves as ordered sequences is beneficial even without changing the underlying filtration.

TOPOFORMER further improves on PH-TR for almost every filtration, typically adding another 1–3 points on most datasets and yielding average gains of 6–7 points over PH-MLP. The effect is especially pronounced on more challenging benchmarks such as IMDB-M and REDDIT-B, where sliding-window interlevel slices capture richer late-emerging structure than standard sublevel PH. Importantly, this pattern holds not only for the three filtrations used in the main text (degree, Ollivier–Ricci, HKS) but also for the four additional ones (betweenness, closeness, eigenvector centrality, Forman–Ricci curvature), indicating that the benefit of Topo-Scan is robust to the choice of scalar function. Together, these results support our central claim: the main performance gains come from the Topo-Scan sequential representation (and its integration with Transformers), rather than from a particular hand-picked filtration function.

## C.8 EARLY SATURATION IN PH FILTRATIONS AND TOPO-SCAN

**Goal.** We compare classical PH (sub/superlevel on a fixed clique-complex 2–skeleton) with *Topo-Scan* to show how PH frequently *early–saturates* on graphs, i.e., after a relatively small portion of the

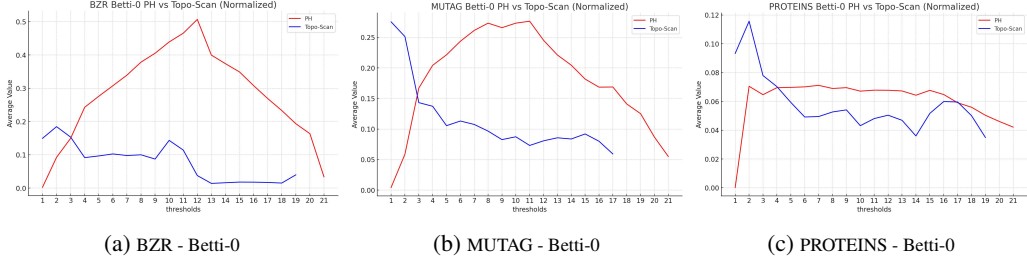

(a) BZR - Betti-0          (b) MUTAG - Betti-0          (c) PROTEINS - Betti-0

Figure 6: **PH vs. Topo-Scan.** Normalized average Betti-0 values over 20 thresholds of the degree-centrality filtration on (a) BZR, (b) MUTAG, and (c) PROTEINS. Under the classical PH pipeline, feature counts decline monotonically with increasing threshold, whereas Topo-Scan maintains elevated values at higher thresholds, revealing late-emerging topological features that PH alone misses.

threshold range, new features cease to appear, whereas Topo-Scan continues to surface structure by sliding windows over the same signal.

**Protocol.** For each dataset and filtration function $f$ (e.g., degree or Ollivier–Ricci), we fix a common grid of $T$ thresholds and evaluate both methods on the same clique-complex 2–skeleton (upper–star from nodes). PH: sublevel filtration evaluated at the same grid points; Betti counts are read at each threshold. Topo-Scan: window width $m$ and stride $s$ define $T$ overlapping slices whose vertex sets correspond to consecutive value ranges in the same grid. Betti counts are computed per slice. To make cross-dataset plots visually comparable, we report (i) *normalized* Betti-0 curves when scales differ markedly (Fig. 6) and (ii) *unnormalized* Betti-0 when PH and Topo-Scan share similar ranges (Fig. 2). Betti-1 frequency barplots are shown to illustrate higher-order behavior (Fig. 7).

**How to read the figures.** A positive Betti-0 value at a position means additional connected components are present in that slice/threshold; persistent nonzero values toward the *right side* of the horizontal axis indicate *late-emerging* structure. For Betti-1, darker bars at higher thresholds indicate more cycles appearing later in the filtration. Because Topo-Scan slices are value–localized *ranges* rather than one-sided sublevels, they retain visibility into regions that are otherwise drowned out once early high- or low-valued nodes saturate the PH complex.

**Results on small biochemical graphs (BZR, MUTAG, PROTEINS).** Figure 6 plots normalized Betti-0 curves over 20 degree thresholds. Across all three datasets, PH curves drop quickly and remain low: after an early rise, new components rarely appear as the complex fills up. In contrast, Topo-Scan maintains elevated values deeper into the axis, indicating that as the sliding window moves, it continues to expose distinct local subgraphs in later value ranges. This pattern is precisely the late-structure retention we aim to capture.

**Results on social graphs (IMDB-B, IMDB-M).** Figure 2 shows unnormalized Betti-0 with 100 thresholds (comparable scales). Here, PH exhibits a sharp taper near the end: once the core of the

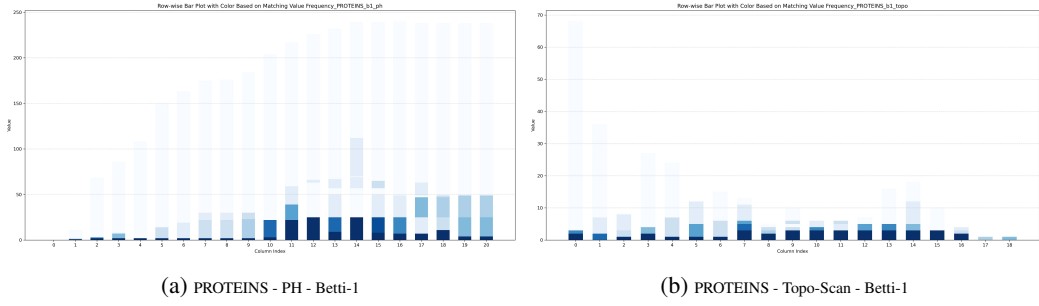

(a) PROTEINS - PH - Betti-1          (b) PROTEINS - Topo-Scan - Betti-1

Figure 7: **PH vs. Topo-Scan.** Bar plots of Betti-1 counts at each O.Ricci filtration threshold on the PROTEINS dataset, with bar color intensity encoding the frequency of each integer value at that threshold. In (a) classical PH features rapidly taper off and plateau early, whereas in (b) Topo-Scan shows increasingly darker bars at higher thresholds, evidence of continued cycle emergence beyond PH's saturation.

graph enters the complex, subsequent thresholds add little. Topo-Scan avoids this collapse; activity persists and often *exceeds* PH in the tail, reflecting components that are still exposed by the windowed slices even when global sublevels have already merged them away.

**Higher-order signal (Betti-1 on PROTEINS, O. Ricci).**    Figure 7 provides barplots where color intensity encodes the frequency of each integer Betti-1 value per threshold. Under PH (left), bars fade and plateau early, showing few cycles after the initial growth phase. Under Topo-Scan (right), darker bars persist across later thresholds, demonstrating continued cycle emergence that PH no longer reveals once the complex has saturated.

**Why does this happen?**    In sub/superlevel PH, once extreme-valued vertices enter early, the induced complex quickly fills in, so later additions create little new topology, especially on graphs where dense regions are correlated with the signal. Topo-Scan, by scanning *ranges* of values with overlap, repeatedly re-centers attention on late parts of the signal, preventing early regions from dominating the entire sequence. Importantly, this is not a claim that sublevel is intrinsically flawed; task-aligned or learned filtrations can mitigate early saturation. Our point is empirical and architectural: a fixed-budget sliding-window view preserves late signal *by design*.

**Controls and caveats.**    (i) We use the same signal, grid, and complex for both methods to avoid confounding factors. (ii) Normalization is applied only for visualization when scales differ; conclusions do not depend on normalization. (iii) Sublevel and superlevel yield the same multiset of slices in reverse order; Topo-Scan's behavior is insensitive to that choice. (iv) Window hyperparameters $(m, s)$ trade locality for coverage; we keep them fixed across datasets in these plots for clarity.

**Takeaway.**    Across biochemical and social benchmarks, PH curves commonly *flatten early*, while Topo-Scan remains *active in the tail* (Betti-0 and Betti-1), revealing late-emerging components and cycles. This supports our central design choice: turning topology into short, ordered, range–localized tokens helps retain information that standard PH pipelines often lose once the complex saturates.

