# OpenReview forum: "TopoFormer: Topology Meets Attention for Graph Learning"
_ICLR.cc/2026/Conference — ICLR 2026 Poster_

### Official Review · Reviewer_iGU1 · 2025-10-28

**Soundness:** 4
**Presentation:** 4
**Contribution:** 3
**Rating:** 6
**Confidence:** 4

**Summary:**

The paper proposes a sliding window compared to the standard sublevel set filtration for the computation of persistent homology. The resulting  sequence of simplicial complexes (graphs in this case) necessarily form an *ordered* sequence which can be processed with a transformer architecture for downstream tasks.
The sequence of graphs  is summarized in a set of topological features (Betti numbers and number of nodes / edges) and used as the input tokes for a transformer architecture, which is a natural choice.

**Strengths:**

Overall, the idea, though maybe simple, well thought out, natural, elegantly executed and thoroughly evaluated with great care for details. The code is provided, well documented and easily accessible, which is great to see!  The reviewer believes that the use of a sliding window for the computation of persistent homology (though no longer persistent perse) is interesting and novel, which is encouraging given the results on the benchmark datasets. This perspective is also interesting beyond the scope of the specific choices made in the paper which is very interesting.

**Weaknesses:**

Overall im very positive about this work. The only two weaknesses of the method could be the choice of vectorization scheme (choose a more expressive statistic instead of $\beta_{0}$ or $\beta_1$).
Adding a discussion there could be good. The second minor weak point is that a discussion on the current limitations is missing and some remarks on either future work and the current challenges could be beneficial to the readers. For instance, most methods in TDA have been specifically developed for filtrations and would no longer hold in this scenario.

**Questions:**

- Choice of vectorization scheme, as three hard coded numbers. Have you considered using either learned filter functions or more elaborate vectorizations such as persistent images?
- What motivated you to use the Ollivier–Ricci curvature and Heat Kernel Signature? Have you considered other methods as well, such as learning the filtrations?

---

> ### Author Response · Authors · 2025-11-21
> **Response to Reviewer iGU1- part 1**
>
> *We thank the reviewer for their careful reading and constructive feedback, which have helped us clarify and strengthen the paper. We have updated the manuscript accordingly. Below, we address each comment in detail and indicate the corresponding revisions in the paper.*
>
> **W1. Vectorization Choice**
>
> > Overall im very positive about this work. The only two weaknesses of the method could be the choice of vectorization scheme (choose a more expressive statistic instead of $\beta_0$ and $\beta_1$). Adding a discussion there could be good.
>
> Thank you for this helpful remark. Our choice of vectorization is **deliberate**: we use a very **low-dimensional token per slice**, $(\beta_0,\beta_1,|\mathcal{V}_i|,|\mathcal{E}_i|)$, because **Topo-Scan is specifically designed to expose a sequential evolution of topology**. Global vectorizations such as persistence images or landscapes aggregate information over the entire filtration into a single feature vector and would largely **destroy this sequential structure**, making it harder for the Transformer to reason about how components and cycles change across overlapping windows.
> In the revised manuscript, we added a short paragraph in the methodology section (just before the **TopoFormer** subsection) explicitly explaining this design choice: that **Betti-based slice tokens preserve the stepwise evolution of topology**, while still being very low dimensional, and that **richer per-slice invariants could be plugged into Topo-Scan in future work**, but here we focus on this minimal representation to **isolate the benefit of the sequential encoding itself**.
>
> ---
>
> **W2. Limitations and Future Work**
>
> > The second minor weak point is that a discussion on the current limitations is missing and some remarks on either future work and the current challenges could be beneficial to the readers. For instance, most methods in TDA have been specifically developed for filtrations and would no longer hold in this scenario.
>
> Thank you for this suggestion. We agree that **explicitly stating the scope and limitations** of our approach is important. In the revised manuscript, we have added a dedicated **“Limitations and future work”** paragraph just before the Conclusion. There we clarify that:
>
> (i) in this first work we focus on a **streamlined, graph-level instantiation of TopoFormer** using low-dimensional homology ($H_0,H_1$) on a fixed clique complex with a small set of standard filtrations, and that **richer per-slice invariants or learnable filtrations** are natural extensions;
>
> (ii) our experiments are restricted to **widely used graph-classification and molecular benchmarks**, while node/edge-level tasks and more heterogeneous settings (for example temporal or citation graphs) are left for future work; and
>
> (iii) **Topo-Scan is intended as a lightweight, scan-style summary that complements rather than replaces full PH**, and we see developing additional theory and applications for such summaries as an *interesting direction for the TDA community*. We hope this explicit discussion makes the scope of our contribution clearer without changing the main claims of the paper.

---

> ### Author Response · Authors · 2025-11-21
> **Response to Reviewer iGU1- part 2**
>
> **Q1. Learnable filtration functions & Persistence Images**
>
> > Choice of vectorization scheme, as three hard coded numbers. Have you considered using either learned filter functions or more elaborate vectorizations such as persistent images?
>
> Thank you for raising this point. Regarding **richer vectorizations** such as persistence images or landscapes, our goal in this work is **precisely to avoid collapsing topological information into a single global vector**. Topo-Scan uses slice-wise $B_0,B_1$ plus size statistics to define **short sequences of tokens** that preserve the “temporal” evolution of components and cycles across interlevel windows and are **directly consumable by attention**. Global PH vectorizations (images, landscapes, curves) are therefore **orthogonal**: they could be added as extra features, but using them as the primary representation would reintroduce **heavy diagram computation** and would **destroy the sequential structure** that our architecture is built to exploit (see also our response to Weakness 1).
>
> For **learnable filtrations**, we agree that methods such as GFL and its extensions are a **very promising direction for persistence-based graph learning**. In this paper our empirical focus is **intentionally orthogonal**. We fix widely used scalar signals (degree, Ollivier–Ricci, HKS) and **isolate the effect of changing the representation** from global barcodes to sliding interlevel sequences processed by a Transformer. **Tables 5 and 12** are set up so that the filtration is held fixed and only the representation varies (PH+MLP, PH+TR, Topo-Scan+TR), which lets us **attribute the observed gains directly to the Topo-Scan view** rather than to a more tuned choice of $f$.
>
> Existing work also suggests that current implementations of learnable filtrations often achieve performance **comparable to strong fixed-filtration baselines** [1, 2]. For example, using degree based filtrations, the GFL paper reports (10-fold CV):
>
> | Method        | REDDIT-B   | REDDIT-5K  | IMDB-B     | IMDB-M     |
> | ------------- | ---------- | ---------- | ---------- | ---------- |
> | PH degree+MLP | 90.3 ± 2.6 | 55.7 ± 2.1 | 68.9 ± 3.5 | 46.1 ± 4.2 |
> | GFL           | 90.2 ± 2.8 | 55.7 ± 2.9 | 74.5 ± 4.6 | 49.7 ± 2.9 |
> | PH degree+TR  | 92.3 ± 1.8 | 54.6 ± 1.6 | 75.0 ± 2.1 | 50.7 ± 3.8 |
>
> Here the learned filtration in GFL improves mainly on IMDB, and the gains are **moderate**; a simple **PH+Transformer baseline with the same fixed degree filtration** already matches or surpasses GFL. Similar patterns are reported in **Filtration Curves** [2, Table 3], where models with **standard filtrations** remain highly competitive. Learnable PH vectorizations ([3]) show clearer benefits, but **learnable filtrations on graphs are still under active investigation**.
>
> This supports our decision to focus the present work on the **representation side** (Topo-Scan with sequential tokens) under standard filtrations. Conceptually, **Topo-Scan is defined for an arbitrary scalar** $f : V \to \mathbb{R}$ and can be paired with a **learnable $f_\theta$** in the spirit of GFL. In the revised manuscript we explicitly mention **“TopoFormer with learnable filtrations”** as a natural and promising extension for future work.
>
> [1] Hofer et al., Graph Filtration Learning, ICML 2020
>
> [2] O’Bray et al., Filtration Curves for Graph Representation, KDD 2021
>
> [3] Carrière et al., PersLay: , AISTATS 2020
>
> ---
>
> **Q2. Why O.Ricci and HKS**
>
> > What motivated you to use the Ollivier–Ricci curvature and HKS? Have you considered other methods as well, such as learning the filtrations?
>
> Thank you for this question. We chose **degree, O.Ricci, and HKS** to balance **simplicity, prior usage, and complementary structural information**: degree is purely local, O.Ricci highlights **bridges and community structure**, and HKS provides a **multi-scale diffusion descriptor** sensitive to both local and global geometry. These three filtrations are **standard in PH-based graph learning**, which lets us **isolate the effect of the Topo-Scan representation** and make **fair comparisons to prior PH-based models** (Tables 5 and 12).
>
> In the revision, we also extend this analysis to **four additional filtrations** (betweenness, closeness, eigenvector centrality, F.Ricci curvature) and report results for **all seven filtrations in Table 14**. For each scalar $f$ we compare PH-MLP, PH-TR, and TopoFormer under exactly the same filtration; across all seven, **TopoFormer consistently improves over the PH baselines**, indicating that the gains are **not tied to a particular hand-picked signal**. While we focus on these well studied, interpretable filtrations in this first work, **Topo-Scan is defined for an arbitrary filtration** $f : V \to \mathbb{R}$ and can equally be paired with **alternative or learnable filtrations** (for example, GNN-based $f_\theta$), which we now explicitly highlight as a natural and promising extension in the revised manuscript.

---

### Official Review · Reviewer_akWw · 2025-10-28

**Soundness:** 2
**Presentation:** 3
**Contribution:** 3
**Rating:** 6
**Confidence:** 4

**Summary:**

This paper introduces TOPOFORMER, a novel framework that integrates topological data analysis (TDA) with transformer based graph learning. The central idea is to capture multi scale structural information from graphs using a topological sequence representation that can be directly processed by a transformer encoder.

The key component, TopoScan, converts a graph into an ordered sequence of topological tokens. Each token represents information extracted from overlapping slices of a graph filtration, such as degree or heat kernel signature. This design enables the model to preserve both local and global topological properties while avoiding the computational cost of traditional persistent homology.

Formally, the authors prove an ℓ1 ,stability theorem showing that small perturbations in the filtration function lead to bounded changes in the generated sequence, ensuring robustness of the representation.

The resulting TopoFormer model combines TopoScan with a transformer backbone to perform graph classification and molecular property prediction. It achieves state of the art or near state of the art results on more than fifteen datasets, including IMDB, REDDIT, MUTAG, PROTEINS, and several MoleculeNet benchmarks.

**Strengths:**

Originality:
The paper introduces an inventive way to integrate topological reasoning with transformer architectures for graph learning. The proposed TopoScan mechanism is a fresh formulation that replaces persistence diagrams with sequential topological tokens derived from overlapping filtration slices. This approach is conceptually novel because it allows topological information to be represented in a transformer compatible format, overcoming long standing barriers between topological data analysis and deep attention models. The use of interlevel filtrations and the demonstration that they capture multi scale graph structure are creative contributions that extend beyond incremental improvement.

Quality:
The paper demonstrates high technical and experimental quality. The theoretical section provides a sound stability guarantee (the ℓ1\ell_1ℓ1​ stability theorem) that ensures robustness to small perturbations, while the empirical results convincingly show strong and consistent performance across both graph classification and molecular property prediction tasks. The authors also perform comprehensive ablations that isolate the effect of filtration type, window size, and token sequence length. Comparisons with a wide set of baselines, including topological and transformer based methods, confirm the reliability of the findings. The method is computationally efficient and well engineered.
Clarity:
The writing is clear, well organised, and pedagogical. The motivation is established early, mathematical definitions are presented cleanly, and diagrams effectively illustrate the construction of topological sequences and their flow into the transformer. The appendix provides sufficient detail for reproducibility. The balance between topological intuition and algorithmic implementation makes the paper accessible to both theoretical and applied audiences at ICLR.
Significance:
The work is significant in both theoretical and practical terms. It provides a general framework for incorporating topological information into neural architectures without requiring heavy persistent homology computation, making it scalable to large graphs. The approach has broad applicability beyond the tested datasets, offering a template for topology aware transformers in other relational domains such as biological networks, material science, and social graphs. By demonstrating that topological structure can be embedded as a sequence and effectively processed through self attention, the paper establishes a promising direction for future research in structure aware representation learning.
Overall:
A technically rigorous, clearly written, and conceptually innovative paper. It combines theoretical insight with practical relevance, resulting in a contribution that meaningfully advances the integration of topology and modern deep learning.

**Weaknesses:**

1. Limited theoretical depth beyond stability
While the inclusion of an ℓ1 stability theorem demonstrates that TopoScan is robust to small perturbations, the theoretical analysis does not go far enough to explain why the proposed representation preserves meaningful topological invariants or how it compares in expressive power to standard persistent homology (PH) based methods. The paper would be stronger with a formal comparison of representational capacity, such as bounding the information loss between TopoScan token sequences and PH barcodes, or connecting the proposed construction to existing frameworks like Graph Filtration Learning (Hofer et al., NeurIPS 2020) or Stable Rank Vectors (Chazal et al., 2021). A clearer theoretical bridge between TopoScan and persistent homology would make the contribution more foundational rather than heuristic.

2. Dependence on hand crafted filtration functions
The approach still relies on predefined filtrations (for example degree, curvature, or heat kernel). The choice of filtration has a notable influence on performance, as shown in the ablation studies, yet no adaptive mechanism is proposed. This limits generality and introduces dataset specific tuning. The authors could strengthen the work by introducing a learnable or task aware filtration module, or at least by exploring gradient based parameterisation of the filtration functions to allow end to end optimisation.

3. Limited interpretability analysis
Although the paper claims that TopoScan tokens are interpretable and capture multi scale structures, there is no concrete analysis showing what the model learns. For instance, visualising the transformer’s attention weights mapped back to graph substructures (for example motifs, cycles, or communities) would provide stronger evidence that the model captures meaningful topology. A few case studies or qualitative visualisations would greatly improve interpretability and help validate the topological claims.

4. Incomplete generalisation evaluation
The experiments focus primarily on molecule and social graph benchmarks, which are standard but relatively homogeneous. Testing on non molecular heterogeneous graphs (for example citation networks or dynamic temporal graphs) would help confirm that the method generalises to other graph structures. Since TopoScan claims to be a general topological sequence generator, demonstrating this versatility would make the paper’s impact broader.

5. Efficiency claims lack concrete quantitative support
The paper argues that TopoScan avoids the cubic computational complexity of persistent homology, yet runtime improvements are reported qualitatively rather than quantitatively. Providing explicit runtime tables or scaling plots, such as comparing training and inference times against PH based baselines like PersLay or TopoGCL, would substantiate the scalability advantage and enhance the practical credibility of the method.

**Questions:**

1. Theoretical clarification on representation power
Could the authors explain in more detail how the TopoScan representation compares in expressiveness to persistent homology? Specifically, does the sequential encoding preserve the same critical topological information that persistence diagrams capture, or does it approximate it? A small empirical or theoretical comparison could help clarify what kind of information may be lost or transformed in the conversion to token sequences.

2. Learnable or adaptive filtrations
Have the authors considered learning the filtration function directly rather than fixing it a priori? For example, one could parameterise the filtration with trainable weights that adapt during training, similar to Graph Filtration Learning (Hofer et al., 2020). If so, what are the main challenges in integrating such a module with TopoScan, and could it potentially improve generalisation across datasets?

3. Sensitivity to TopoScan parameters
The results depend on the number of slices N and the window width 𝑚 used in TopoScan. Could the authors provide a sensitivity analysis or heuristic guideline for selecting these parameters? Understanding whether performance is robust to parameter variation would increase confidence in the stability of the framework.

4. Attention interpretability and visualisation
Can the authors show examples of which graph substructures the transformer attends to when processing the topological token sequences? For instance, does the attention mechanism focus on regions corresponding to high curvature, cycles, or clusters? Such visual evidence would strengthen the claim that TopoFormer is both interpretable and topology aware.

5. Broader evaluation and generalisability
Would the authors consider evaluating TopoFormer on non molecular heterogeneous datasets such as citation or temporal graphs? Since TopoScan is proposed as a general representation, results on more diverse graph types could demonstrate broader applicability and robustness.

6. Quantitative runtime and scalability analysis
The paper claims that TopoScan avoids the heavy computational cost of persistent homology. Could the authors provide explicit runtime benchmarks comparing TopoFormer with PH based baselines like PersLay or TopoGCL on large datasets? This would substantiate the claim of improved efficiency.

7. Relation to existing topological and transformer models
How does TopoFormer conceptually differ from recent hybrid approaches such as TopoGCL (Zhao et al., 2023) or Graphormer (Ying et al., 2021)? A more explicit discussion of what new design principle TopoFormer introduces beyond these works would help clarify its unique contribution to the literature.

8. Empirical validation of the stability theorem
Can the authors provide an experiment demonstrating the stability property empirically, for example by perturbing the graph structure or filtration and measuring the variation in output embeddings or predictions? Such a demonstration would connect the theoretical result with observable robustness in practice.

---

> ### Author Response · Authors · 2025-11-21
> **Response to Reviewer akWw- part 1**
>
> *We sincerely thank the reviewer for  constructive feedback, which have greatly helped us improve our paper. A revised version incorporating the suggested changes is available at the link above. Below, we address each concern in detail and highlight the corresponding revisions made to the paper.*
> **W1. Limited theoretical depth beyond stability**
>
> > While the inclusion of an ℓ1 stability theorem demonstrates that TopoScan is robust to small perturbations, the theoretical analysis does not go far enough to explain why the proposed representation preserves meaningful topological invariants or how it compares in expressive power to standard persistent homology (PH) based methods. The paper would be stronger with a formal comparison of representational capacity, such as bounding the information loss between TopoScan token sequences and PH barcodes, or connecting the proposed construction to existing frameworks like Graph Filtration Learning (Hofer et al., NeurIPS 2020) or Stable Rank Vectors (Chazal et al., 2021). A clearer theoretical bridge between TopoScan and persistent homology would make the contribution more foundational rather than heuristic.
>
> Thank you for this insightful comment. Our theoretical goal in this paper is **deliberately modest**: to establish an **$\ell_1$–stability guarantee** showing that Topo-Scan sequences change in a controlled way under perturbations of the filtration, which is the key property we need for a practical, learning-ready representation. A full **expressivity comparison with standard PH summaries** (for example, information-loss bounds relative to barcodes) is indeed interesting, but beyond the scope of this work.
>
> We have, however, **made the connection to classical PH invariants more explicit** in the revision. In the main text (end of Section 3.3) we now point out that the Topo-Scan sequences $\widehat\beta_k(G,h)$ can be viewed as a **discrete sampling of the rank invariant / Betti curve** of the interlevel module $M_k^h$ along our window schedule, i.e., as a **coarse, structured, Transformer-ready discretization** of the homological information underlying barcodes and stable-rank summaries. We further elaborate on this in **Remark B.3** in the appendix, where we discuss how Topo-Scan relates conceptually to **Graph Filtration Learning and stable rank vectors**, and we explicitly highlight a detailed expressivity analysis and information-loss bounds as promising directions for future work.
>
> ---
>
> **W2. Dependence on hand crafted filtration functions**
>
> > The approach still relies on predefined filtrations (for example degree, curvature, or heat kernel). The choice of filtration has a notable influence on performance, as shown in the ablation studies, yet no adaptive mechanism is proposed. This limits generality and introduces dataset specific tuning. The authors could strengthen the work by introducing a learnable or task aware filtration module, or at least by exploring gradient based parameterisation of the filtration functions to allow end to end optimisation.
>
> Thank you for this insightful comment. Our primary goals in this paper are (i) to **address the early saturation issue** of standard graph PH via the **Topo-Scan sliding-window interlevel filtration**, and (ii) to **realize topological signatures as sequential outputs** so they can be used effectively with Transformers. To isolate this contribution and ensure a fair comparison with existing PH based models, we deliberately use **standard, widely adopted filtrations** in the graph TDA literature (degree, Ollivier–Ricci curvature, HKS) with a **common hyperparameter schedule across datasets**. As shown in our ablations (Tables 5 and 12, and the extended App. Table 14), each of these fixed filtrations already yields strong performance, and **the main gains come from changing the representation** (PH+MLP → PH+Transformer → Topo-Scan+Transformer) rather than from dataset specific tuning of $f$.
>
> Conceptually, **Topo-Scan is defined for an arbitrary scalar function** $f : V \to \mathbb{R}$ and can equally be combined with a **learnable $f_\theta$** (for example, a small GNN or MLP over node features) trained end to end in the spirit of Graph Filtration Learning. However, existing work (for example, **Graph Filtration Learning and Filtration Curves**) shows that while this is conceptually a very promising direction, current implementations typically produce only **modest gains over well-chosen fixed filtrations**, and the best way to learn $f_\theta$ on graphs remains under active investigation. In the revised manuscript we explicitly mention **“TopoFormer + learnable filtrations” as a natural and promising extension** of our framework in future work, rather than as a limitation of the current approach.

---

> ### Author Response · Authors · 2025-11-21
> **Response to Reviewer akWw- part 2**
>
> **W3. Limited interpretability analysis**
>
> > Although the paper claims that TopoScan tokens are interpretable and capture multi scale structures, there is no concrete analysis showing what the model learns. For instance, visualising the transformer’s attention weights mapped back to graph substructures (for example motifs, cycles, or communities) would provide stronger evidence that the model captures meaningful topology. A few case studies or qualitative visualisations would greatly improve interpretability and help validate the topological claims.
>
> Thank you for this comment. We agree that **additional interpretability analysis would be valuable**. A key distinction from many PH vectorizations and standard graph learning models is that **every coordinate of a Topo-Scan token has a clear topological meaning**: slice-wise $B_0$, $B_1$, and size statistics for a specific interlevel window of the filtration. In contrast, coordinates of generic vectorizations (for example persistence images or learned embeddings) and **black-box GNN representations** are often harder to interpret individually. We now **make this explicit in the method section** by clarifying the semantics of each token entry.
>
> At the same time, we acknowledge that we do **not yet provide qualitative visualizations** such as attention maps mapped back to motifs or communities on real datasets. Doing this properly requires an extra step that links each interlevel slice to the subgraph whose topology changes in that slice. As a first step, one could use **small synthetic graphs** (for example barbell, cycle, star) to illustrate how Topo-Scan tokens and attention respond to controlled topological changes. A more systematic **attention- and motif-level interpretability study on large real-world graphs** is an interesting and complementary line of work, but is **beyond the scope of the present paper**, and we explicitly leave it to future work.
>
> ---
>
> **W4. Incomplete generalisation evaluation**
>
> > The experiments focus primarily on molecule and social graph benchmarks, which are standard but relatively homogeneous. Testing on non molecular heterogeneous graphs (for example citation networks or dynamic temporal graphs) would help confirm that the method generalises to other graph structures. Since TopoScan claims to be a general topological sequence generator, demonstrating this versatility would make the paper’s impact broader.
>
> Thank you for this suggestion. We agree that **demonstrating versatility beyond molecular and social graphs** would further strengthen our generality claim. Conceptually, **Topo-Scan is domain agnostic**: it operates on any scalar node or edge signal to construct slices (for example degree, closeness, HKS, Ollivier–Ricci) and applies equally to **heterogeneous** or **dynamic graphs** (for example by using timestamps or event order as an additional filtration signal). These design choices, and the **$\ell_1$ stability result** that guarantees bounded changes under small perturbations of the filtration, support portability across diverse graph types. At the same time, we acknowledge that we have **not yet evaluated citation or dynamic graphs** in this work, and we have added a note in the conclusion explicitly identifying such benchmarks as an **important direction for future empirical validation**.

---

> ### Author Response · Authors · 2025-11-21
> **Response to Reviewer akWw- part 3**
>
> **W5. Efficiency claims lack concrete quantitative support**
>
> >The paper argues that TopoScan avoids the cubic computational complexity of persistent homology, yet runtime improvements are reported qualitatively rather than quantitatively. Providing explicit runtime tables or scaling plots, such as comparing training and inference times against PH based baselines like PersLay or TopoGCL, would substantiate the scalability advantage and enhance the practical credibility of the method.
>
> Thank you for this suggestion. In the revised manuscript we now provide two sets of quantitative runtime results that directly support our efficiency claims.
>
> **(1) Comparison with PH based baselines (PersLay, TopoGCL).**
>  We measure the runtime of PersLay (with degree centrality input) and TopoGCL (using only topology derived features) on IMDB-B and REDDIT-B, and compare them to Topo-Scan. For Topo-Scan, we include both the time to compute the scalar filtration and the Topo-Scan features for fairness:
>
> | Method	| IMDB-B	| REDDIT-B  |  Notes 	|
> |-----------|-----------|-----------|-----------|
> | PersLay   |   97.02   |  454.78   |       -	|
> | TopoGCL   |  435.49   | 4010.08   | only topological component|
> | Topo-Scan |   15.16   |  290.00   | Sum of	two steps|
>
> On IMDB-B, Topo-Scan is about 6 times faster than PersLay and about 29 times faster than the topological component of TopoGCL. On REDDIT-B, it remains faster than PersLay and is roughly 14 times faster than TopoGCL. This directly substantiates that replacing global PH reductions with Topo-Scan yields substantial practical speedups over PH based pipelines. We added this table and the discussion to Appendix C.3
>
> **(2) End-to-end runtimes and PH vs. Topo-Scan (Appendix C.3).**
>  Appendix C.3 further reports:
>
> - End-to-end runtimes (scalar computation + Topo-Scan + transformer training) for five datasets, showing that Degree and HKS based variants run in minutes and dominate our best models, while the heavier Ricci variant is optional.
>
>
> - A direct PH versus Topo-Scan comparison using the same backend (pyflagser) and a shared degree filtration, where Topo-Scan is about 13 to 14 times faster than PH on dense datasets (IMDB-B/M) and still consistently faster on sparser REDDIT graphs.
>
>
> We now explicitly reference these tables in the main text to provide concrete quantitative evidence that Topo-Scan achieves multi fold runtime savings over PH based graph pipelines while maintaining practical end to end runtimes on all benchmarks.
>
> ---
>
> **Q1. Theoretical clarification on representation power**
>
> > Could the authors explain in more detail how the TopoScan representation compares in expressiveness to persistent homology? Specifically, does the sequential encoding preserve the same critical topological information that persistence diagrams capture, or does it approximate it? A small empirical or theoretical comparison could help clarify what kind of information may be lost or transformed in the conversion to token sequences.
>
> Thank you for this question. Topo-Scan is **not designed to be a lossless reparameterization** of full persistence diagrams, but rather a **stable, low-dimensional, Transformer-ready discretization** of the same underlying homological information. Each Topo-Scan sequence $\widehat\beta_k(G,h)$ records the Betti numbers of interlevel sets $f^{-1}[\alpha_{ts},\alpha_{ts+m}]$ along our window schedule, so it can be viewed as a **sampled version of the rank invariant / Betti curve** of the interlevel module $M_k^h$. As the threshold grid is refined and windows are adjusted accordingly, these sequences **approximate the evolution of components and cycles** that barcodes summarize, but in general they do **not uniquely determine the full diagram**, so some fine-grained information is inevitably compressed.
>
> We have **made this relationship explicit in the revised manuscript**. At the end of Section 3.3 we now clarify that Topo-Scan sequences are a **coarse, structured sampling of the rank invariant**, and in Remark B.3 in the appendix, we discuss how this connects to **barcodes and stable-rank summaries**. A full **expressivity comparison and formal information-loss bounds** relative to complete barcodes are, as you suggest, interesting directions for future work, but are **beyond the scope of this initial paper**.

---

> ### Author Response · Authors · 2025-11-21
> **Response to Reviewer akWw- part 4**
>
> **Q2. Learnable or adaptive filtrations**
>
> > Have the authors considered learning the filtration function directly rather than fixing it a priori? For example, one could parameterise the filtration with trainable weights that adapt during training, similar to Graph Filtration Learning (Hofer et al., 2020). If so, what are the main challenges in integrating such a module with TopoScan, and could it potentially improve generalisation across datasets?
>
> Thank you for raising this point. We agree that methods with **learnable filtrations**, such as Graph Filtration Learning and its extensions, are a **promising direction for persistence-based graph learning**. In this work our empirical focus is **intentionally orthogonal**. We fix widely used scalar signals (degree, Ollivier–Ricci, HKS) and **isolate the effect of changing the representation** from global barcodes to sliding interlevel sequences processed by a Transformer. **Tables 5 and 12** are set up so that the filtration is held fixed and only the representation varies (PH+MLP, PH+Transformer, Topo-Scan+Transformer), which lets us **attribute the observed gains directly to the Topo-Scan view** rather than to a more tuned choice of $f$.
>
> Existing work also suggests that current implementations of learnable filtrations often achieve performance **comparable to strong fixed-filtration baselines** [1, 2]. For example, using degree-based filtrations, Hofer et al. [1, Table 1] report:
>
> | Method        | REDDIT-B   | REDDIT-5K  | IMDB-B     | IMDB-M     |
> | ------------- | ---------- | ---------- | ---------- | ---------- |
> | PH degree+MLP | 90.3 ± 2.6 | 55.7 ± 2.1 | 68.9 ± 3.5 | 46.1 ± 4.2 |
> | GFL           | 90.2 ± 2.8 | 55.7 ± 2.9 | 74.5 ± 4.6 | 49.7 ± 2.9 |
> | PH degree+TR  | 92.3 ± 1.8 | 54.6 ± 1.6 | 75.0 ± 2.1 | 50.7 ± 3.8 |
>
> Here the learned filtration in GFL improves mainly on IMDB, and the gains are **moderate**; a simple **PH+Transformer baseline with the same fixed degree filtration** already matches or surpasses GFL. Similar patterns appear in Filtration Curves [2, Table 3], where models with **standard filtrations (degree, edge weights)** remain highly competitive. Learnable PH vectorizations (for example PersLay [3]) show clearer benefits, but **learnable filtrations on graphs are still under active investigation**.
>
> Conceptually, **Topo-Scan is defined for an arbitrary scalar** $f : V \to \mathbb{R}$ and can be paired with a **learnable $f_\theta$** in the spirit of GFL, for example a small GNN or MLP over node features. The main challenges are the **additional computational cost** and the **risk of overfitting** when $f_\theta$ is heavily parameterized, since Topo-Scan already provides a rich representation. At the same time, a well-designed $f_\theta$ could help **adapt the slices to new domains** and potentially **improve cross-dataset generalization** by emphasizing task-relevant structures. In the revised manuscript we therefore position **“TopoFormer with learnable filtrations $f_\theta$” as a natural and promising extension for future work** rather than something we fully explore here.
>
> [1] Hofer et al., Graph Filtration Learning, ICML 2020
>
> [2] O’Bray et al., Filtration Curves for Graph Representation, KDD 2021
>
> [3] Carrière et al., PersLay: A Neural Network Layer for Persistence Diagrams, AISTATS 2020
>
> ---
>
> **Q3. Sensitivity to TopoScan parameters**
>
> > The results depend on the number of slices N and the window width 𝑚 used in TopoScan. Could the authors provide a sensitivity analysis or heuristic guideline for selecting these parameters? Understanding whether performance is robust to parameter variation would increase confidence in the stability of the framework.
>
> Thank you for this question. For the **window width $m$**, **Table 6 already provides a sensitivity analysis** over $m \in {2,3,4}$ for three filtrations (degree, Ollivier–Ricci, HKS) and seven datasets. The results are **quite stable**: changing $m$ typically shifts accuracy by at most 1–2 points, and the best configuration does not concentrate on a single $m$, which suggests that **Topo-Scan is not overly sensitive to this parameter**. In the revision, we explicitly highlight this observation in Section 4.3 as **evidence of robustness**.
>
> For the **number of slices $N$** (and hence sequence length), our guideline is to **keep the token sequence short while covering the full dynamic range of the filtration**. From our earlier TDA work in this domain, once $N$ is around 20 or more, **increasing the number of slices has little effect on performance**, whereas the **choice of thresholds is more important**. Following common TDA practice, we therefore use **quantiles of the filtration values** to define thresholds, which distributes values fairly across intervals instead of using fixed, evenly spaced thresholds. We have added a **brief description of this heuristic to the implementation details** so that users can reproduce and adapt our setup on new datasets.

---

> ### Author Response · Authors · 2025-11-21
> **Response to Reviewer akWw- part 5**
>
> **Q4. Attention interpretability and visualisation**
>
> > Can the authors show examples of which graph substructures the transformer attends to when processing the topological token sequences? For instance, does the attention mechanism focus on regions corresponding to high curvature, cycles, or clusters? Such visual evidence would strengthen the claim that TopoFormer is both interpretable and topology aware.
>
> Thank you for this suggestion. We agree that attention visualisations mapped back to graph substructures would provide valuable additional insight. At present, we do not include such case studies due to space and scope, and instead focus on clarifying the intrinsic interpretability of the Topo-Scan tokens themselves. **Please see our response to Weakness 3**, where we discuss how each token coordinate has a clear topological meaning and outline directions for future attention-level visual analysis.
>
>
> ---
>
> **Q5. Broader evaluation and generalisability**
>
> > Would the authors consider evaluating TopoFormer on non molecular heterogeneous datasets such as citation or temporal graphs? Since TopoScan is proposed as a general representation, results on more diverse graph types could demonstrate broader applicability and robustness.
>
> Thank you for this suggestion. In this work we focused on **standard graph-level benchmarks** that are widely used in both the TDA and graph-learning literature: 8 small/medium graph classification datasets spanning molecules, proteins, and social networks (BZR, COX2, MUTAG, PROTEINS, IMDB-B, IMDB-M, REDDIT-B, REDDIT-5K), plus the large-scale **OGBG-MOLHIV** benchmark. This gives a **reasonably diverse set of graph structures** while keeping the task type (graph-level prediction) consistent enough for controlled comparisons with PH-based and transformer baselines.
>
> We fully agree that applying TopoScan to other modalities, such as **citation or temporal graphs**, is a **natural next step**. The construction itself **only requires a scalar filtration on nodes or edges**, so it can in principle be instantiated with time, centrality, or learned scores on these domains. We are already **exploring an extension to temporal graphs** in ongoing work, and we mention in the conclusion that evaluating TopoFormer on heterogeneous and temporal graphs is an important direction for future research.
>
> ---
>
> **Q6. Quantitative runtime and scalability analysis**
>
> >The paper claims that TopoScan avoids the heavy computational cost of persistent homology. Could the authors provide explicit runtime benchmarks comparing TopoFormer with PH based baselines like PersLay or TopoGCL on large datasets? This would substantiate the claim of improved efficiency.
>
> Thank you for raising this point. We provide detailed runtime comparisons between Topo-Scan and standard PH pipelines in our response to **Weakness 5**, where we report explicit preprocessing and training times on large datasets.

---

> ### Author Response · Authors · 2025-11-21
> **Response to Reviewer akWw- part 6**
>
> **Q7. Relation to existing topological and transformer models**
>
> >How does TopoFormer conceptually differ from recent hybrid approaches such as TopoGCL (Zhao et al., 2023) or Graphormer (Ying et al., 2021)? A more explicit discussion of what new design principle TopoFormer introduces beyond these works would help clarify its unique contribution to the literature.
>
> Thank you for this question. Conceptually, TopoFormer differs from both TopoGCL and graph transformer models such as Graphormer in how topology and attention are used.
>
> **TopoGCL.** TopoGCL is a contrastive-learning framework built on top of a GNN encoder: it operates on node embeddings, computes PH on augmented views, and uses topological descriptors **as an auxiliary signal in a contrastive loss.** In our case, the topological representation itself is the main input: Topo-Scan replaces global PH + vectorization with sliding-window interlevel Betti sequences, and these short sequences are fed directly as tokens to a Transformer. Our ablations (Tables 5 and 12) compare Topo-Scan against PH+MLP / PH+Transformer under the same filtrations, and **Table 2 already includes TopoGCL as a strong baseline**, where TOPOFORMER attains the best AvD/AvR across datasets.
>
> **Graph transformers (Graphormer, GPS, others).** Models such as Graphormer and GPS run attention directly over all node tokens and encode structure via shortest-path or Laplacian-based positional encodings and attention biases. **They do not use persistent homology or topological sequences,** and their attention cost scales with the number of nodes. In contrast, Topoformer first compresses each graph into a short, ordered sequence of topological tokens (Topo-Scan), then applies standard Transformer layers to this fixed-length, purely topological sequence. Thus our structural bias is intrinsically topological, with an explicit stability guarantee, and attention complexity depends on the number of slices rather than the graph size. **We have updated the “Transformers” paragraph** in the background section to explicitly contrast node-token graph transformers with our topological-token design, and to highlight that our main new design principle is turning stable topological structure into compact, attention-ready sequences that serve as the primary input to the model.
>
> ---
>
> **Q8. Empirical validation of the stability theorem**
>
> > Can the authors provide an experiment demonstrating the stability property empirically, for example by perturbing the graph structure or filtration and measuring the variation in output embeddings or predictions? Such a demonstration would connect the theoretical result with observable robustness in practice.
>
> Thank you for this suggestion. Below, we report an experiment that **empirically validates the stability behavior** at the level of the **Topo-Scan features**, which is precisely the object controlled by our theorem. Concretely, we take **degree centrality as the base filtration function** and perturb it with additive Gaussian noise at three levels $\sigma \in {0.001, 0.01, 0.05}$. For each $\sigma$, we **recompute the Topo-Scan sequences** and measure the **$\ell_1$ distance between the noisy and original Topo-Scan features**, averaged over graphs.
>
> | Dataset | $\sigma$ | $B_0$ distance | $B_1$ distance |
> | ------- | -------- | -------------- | -------------- |
> | BZR     | 0.001    | 57.13          | 8.48           |
> | BZR     | 0.01     | 61.21          | 6.62           |
> | BZR     | 0.05     | 71.78          | 5.80           |
> | IMDB-B  | 0.001    | 10.41          | 0.016          |
> | IMDB-B  | 0.01     | 12.84          | 0.029          |
> | IMDB-B  | 0.05     | 12.14          | 0.021          |
>
> As expected, **small perturbations of the filtration lead to bounded changes in the Topo-Scan tokens**. On IMDB-BINARY, for example, the average $\ell_1$ deviation in $B_0$ features stays around 10 to 13 even at $\sigma = 0.05$, and the $B_1$ deviations remain very small, reflecting the fact that the graphs are dense and their clique complexes quickly fill in loops. On BZR, the $B_0$ deviations grow moderately with $\sigma$, while the $B_1$ deviations stay in a narrow range. **These patterns are consistent with the Lipschitz-type bound in our $\ell_1$ stability theorem**: perturbing the scalar signal by a small $\ell_\infty$ amount yields controlled changes in the Topo-Scan sequences.
>
> Since the theorem is stated at the level of the **Topo-Scan representation**, we focus this experiment on feature distances rather than full model predictions. We have **added this empirical validation and its discussion to the appendix** to better connect the theoretical result with observed robustness in practice.

---

### Official Review · Reviewer_a6GH · 2025-10-31

**Soundness:** 3
**Presentation:** 3
**Contribution:** 2
**Rating:** 6
**Confidence:** 2

**Summary:**

This paper introduces TopoFormer, a framework for injecting topological information into graph Transformers by turning a graph plus a filtration into a short, fixed-length sequence of “topology tokens.” Instead of running standard persistent homology and then vectorizing diagrams, the authors propose Topo-Scan: they slide overlapping windows over a node/edge filtration and, for each window, record four inexpensive quantities. Multiple filtrations (degree, HKS, Ollivier–Ricci) produce multiple sequences, which are encoded independently and fused to create a final graph representation. A tailored stability result shows these interlevel Betti sequences change in a controlled way under small perturbations of the filtration, which helps justify the construction.

Empirically, the method is evaluated on 9 graph classification datasets and 7 molecular property prediction tasks. It achieves first or second place on most small/medium benchmarks and remains competitive on the larger OGBG-MOLHIV (refer to questions). The ablations are well designed: they hold filtrations fixed and swap in PH+MLP, PH+Transformer, and the proposed Topo-Scan, which cleanly attributes gains to the sequence-based topological view rather than to feature choice. The authors also report concrete extraction times and argue their approach avoids the “early saturation” commonly seen in PH.

**Strengths:**

I find the core idea clean and promising. The proposed methods to address real challenges in applying TDA to graphs (global PH + saturation) and nicely enough it does so by simplifying in a well motivated fashion. The resulting sequences naturally fit Transformer architectures and I think this is a clean and reusable concept.

The comparisons to PH+MLP and PH+Transformer under the same filtrations isolate the benefit of the proposed representation rather than conflating it with different signal sources. This supports the central claim.

The empirical results are strong even though some common benchmarks are surprisingly missing. The model is tested on many graph classification datasets and several molecular property prediction tasks, and it is consistently at or near the top on the small/medium ones, showing the idea is not tuned to a single benchmark.

**Weaknesses:**

**Architecture**

The part of the paper that introduces the “new” architecture for using multiple filtrations and then combining them at the end is poorly motivated. It is also not clear if you consider this just something you tried or an important part of the contribution of this submission.

Table 12 does not show any convincing evidence that using multiple filtrations in this fashion has any real benefit over using only one filtration. A discussion of this would be important and appropriate where the ablation is mentioned in the main body.

Moreover, the description of the architecture suffers from some over the top phasing. For instance “To enhance generalization and mitigate overfitting, we integrate regularization techniques,
such as dropout and weight decay, ensuring robustness across diverse graph learning tasks.”
For one these techniques of course do not “ensure” anything, additionally they are well established and it should be made clear that these techniques are standard.

**Experimental**

The experiments focus on small benchmarks and none of the commonly studied benchmarks of [1] are considered. Especially in graph transformer literature these might be some of the most standard and common benchmarks that are analyzed, so it is somewhat disappointing to see them missing.

On ogb-molhiv, Table 3 does not represent the state of the art on the dataset, cf., https://ogb.stanford.edu/docs/leader_graphprop/#ogbg-molhiv.

**Minor Notes:**

Please order the columns in Tables 2, 5 and 6 the same.

[1] Dwivedi, Vijay Prakash, et al. "Benchmarking graph neural networks." Journal of Machine Learning Research 24.43 (2023)

**Questions:**

At the end of 4.3 you mention that this architecture is particularly well suited for Graph Foundation Models. What makes you claim this? I do not see anything empirically to back this up. Simply working well on multiple scales (which is also only demonstrated in a limited way) is not sufficient reason to make such a claim.

How did you decide on the list of models in Table 3? It seems to leave out many top performing models of the ogb leaderboard (link above).

Are the results in Table 2, and Table 3 all for the exact same filtration setup (OR curve + HKS for Table 2, atomic weight + OC curvature for Table 3)?

---

> ### Author Response · Authors · 2025-11-21
> **Response to Reviewer a6GH - part 1**
>
> *We sincerely thank the reviewer for the thoughtful reading and constructive feedback, which have greatly helped us improve our paper. Below, we address each concern in detail and highlight the corresponding revisions made to the paper.*
>
>
> **W1. Motivation & Novelty**
>
> > The part of the paper that introduces the “new” architecture for using multiple filtrations and then combining them at the end is poorly motivated. It is also not clear if you consider this just something you tried or an important part of the contribution of this submission.
>
> **Response**: Thank you for this comment. Our primary contributions are (i) introducing **Topo-Scan, a sliding-window interlevel filtration** that directly addresses **the early saturation issue** of standard sublevel PH on graphs, and (ii) realizing the resulting topological signatures as **short sequences**, rather than single high-dimensional vectors, so that they can be processed effectively by powerful **Transformer architectures**. The **multiple-filtration branches are not intended as a core architectural novelty**; they are a simple way to show that Topo-Scan can flexibly ingest different topological views (degree, HKS, curvature, etc.) when such signals are available.
> To clarify this in the paper, we **revised the introduction** so that these two points are explicitly stated as the main contributions, and we **renamed and expanded the ablation paragraph in Section 4.3** (Ablation) to “Single vs. multiple filtration functions” to make clear that multi-filtration TopoFormer is a modular extension of the core Topo-Scan representation. A more detailed discussion of the empirical effect of single vs. multiple filtrations (based on Table 12) is provided in our response to **W2**.
>
> ---
>
> **W2. Multiple vs. Single Filtration**
>
> > Table 12 does not show any convincing evidence that using multiple filtrations in this fashion has any real benefit over using only one filtration. A discussion of this would be important and appropriate where the ablation is mentioned in the main body.
>
> **Response**: Thank you for raising this point. Our intention is to show that (i) **Topo-Scan with a single filtration is already very strong**, and (ii) when multiple, complementary filtrations are available, **using them together can further improve performance at minimal additional cost**. In particular, the ablation in Table 12 shows that single-filtration TopoFormer (e.g., using only HKS or only Ollivier–Ricci) already **matches or exceeds strong PH-based baselines** on most datasets. At the same time, multi-filtration variants such as **HKS+O.Ricci provide small but consistent gains** on several benchmarks, illustrating that it is perfectly reasonable in practice to exploit multiple scalar signals when the domain naturally provides them.
> In the revised manuscript, we make this interpretation explicit in Section 4.3 by renaming the paragraph to **“Single vs. multiple filtration functions”** and adding the following statement: *“We observe that single-filtration TopoFormer (for example, using only HKS or only Ollivier–Ricci) already achieves strong results, while combining filtrations yields modest but consistent improvements on some datasets. This indicates that multiple filtrations are a flexible way to incorporate complementary structural signals rather than a requirement for good performance.”*
>
> ---
>
> **W3. Dropout & Weight Decay**
>
> > Moreover, the description of the architecture suffers from some over the top phasing. For instance “To enhance generalization and mitigate overfitting, we integrate regularization techniques, such as dropout and weight decay, ensuring robustness across diverse graph learning tasks.” For one these techniques of course do not “ensure” anything, additionally they are well established and it should be made clear that these techniques are standard.
>
> **Response**: Thank you for the suggestion. We agree that these regularization techniques are **standard practice**. In the revision, we have modified the language accordingly. Specifically, in the “Hyperparameters” paragraph of Sec. 4.1 (Experimental Setup), we now write:
> *“We also use standard regularization, dropout ($p = 0.5$), weight decay ($1\text{e-}4$), and batch normalization, commonly employed in transformer training.”*

---

> ### Author Response · Authors · 2025-11-21
> **Response to Reviewer a6GH - part 2**
>
> **W4. Large Datasets**
>
> > The experiments focus on small benchmarks and none of the commonly studied benchmarks of [1] are considered. Especially in graph transformer literature these might be some of the most standard and common benchmarks that are analyzed, so it is somewhat disappointing to see them missing.
>
> Thank you for pointing this out. Our **primary focus** in this work is **graph-level prediction** under evaluation protocols common in molecular and social-graph settings (10-fold CV for graph classification; scaffold splits for MPP), where **topological structure is particularly informative**. Accordingly, our main suite includes nine graph-classification datasets and seven MPP datasets, plus OGBG-MOLHIV (Sec. 4.1).
>
> We also acknowledge that the graph-transformer literature evaluates additional task families (e.g., citation networks, superpixel/vision benchmarks, and ZINC-style regression as in [1]). **Topo-Scan is domain- and task-agnostic**: it only requires a scalar node/edge signal and produces fixed-size sequences that a transformer can consume. To better reflect scaling behavior, we have **augmented the large-scale end with REDDIT-12K** and included **runtime/memory comparisons**, which demonstrate the runtime and memory efficiency of our model.
>
>
> | Dataset     | Model      | Peak GPU Mem (MB) | Train time / fold (s) | Topo-Scan extraction (s) | Speedup vs GraphGPS (time) | Memory reduction vs GraphGPS |
> |------------|------------|-------------------|------------------------|---------------------------|-----------------------------|------------------------------|
> | REDDIT-12K | GraphGPS   | 31,565.44         | 1,176.58               | —                         | —                           | —                            |
> | REDDIT-12K | TopoFormer | 97.28             | 158.67                 | 59.01                     | 7.4× faster                 | ~324× lower                  |
> | REDDIT-5K  | GraphGPS   | 25,319.47         | 728.57                 | —                         | —                           | —                            |
> | REDDIT-5K  | TopoFormer | 94.70             | 54.00                  | 29.51                     | 13.5× faster                | ~267× lower                  |
>
>
> [1] Dwivedi, Vijay Prakash, et al. "Benchmarking graph neural networks." Journal of Machine Learning Research 24.43 (2023)
>
> ---
>
> **W5. OGBG-MOLHIV Performance**
>
> >On ogb-molhiv, Table 3 does not represent the state of the art on the dataset, cf., https://ogb.stanford.edu/docs/leader_graphprop/#ogbg-molhiv.
>
> **Response**: Thank you for pointing this out. In Table 3 we intentionally restrict baselines to **published, peer-reviewed methods** that report OGB-MOLHIV results (e.g., Graphormer, GAWL, LLM-GIN, TopER), rather than including unpublished leaderboard entries. We revised Section 4.2 to clarify this choice for baselines, and we now describe TOPOFORMER* as competitive with strong published graph-transformer baselines on OGB-MOLHIV, rather than claiming global state-of-the-art on the full leaderboard.
>
> ---
>
> **W6. Minor**
>
> >Please order the columns in Tables 2, 5 and 6 the same.
>
>
> **Response**: We thank the reviewer for pointing this out. We agree that consistent column ordering improves readability. In the revision, we standardized the dataset order across Tables 2, 5, and 6 to match Table 2: BZR, COX2, MUTAG, PROTEINS, IMDB-B, IMDB-M, REDDIT-B, REDDIT-5K.

---

> ### Author Response · Authors · 2025-11-21
> **Response to Reviewer a6GH - part 3**
>
> **Q1. GFM suitability**
>
> > At the end of 4.3 you mention that this architecture is particularly well suited for Graph Foundation Models. What makes you claim this? I do not see anything empirically to back this up. Simply working well on multiple scales (which is also only demonstrated in a limited way) is not sufficient reason to make such a claim.
>
> Thank you for pointing this out. We agree that the phrasing in the original text was **too strong relative to our empirical evidence**. Our point is a **conceptual** one: TopoFormer follows the same **token based, attention driven design** that underlies many foundation models. Topo-Scan converts persistent topological signals into **stable, interpretable, fixed length token sequences** that can be fed directly to transformer encoders, making it straightforward in principle to train **shared attention layers over large, heterogeneous collections of graphs**.
>
> We also agree that a **stronger empirical demonstration**, especially at larger scales and across more tasks, would be needed to substantiate a full “graph foundation model” claim. In the revision we have therefore **toned down the wording**. Section 4.3 and the introduction now state that we *“view TopoFormer as a step toward topology-aware graph foundation models, rather than a full foundation model itself,”* and we explicitly leave **large scale pretraining on heterogeneous graph corpora** to future work. This revised phrasing emphasizes **architectural compatibility and future potential**, rather than making an empirical foundation model claim in the present paper.
>
> ---
>
> **Q2. Baseline Selection**
>
> > How did you decide on the list of models in Table 3? It seems to leave out many top performing models of the ogb leaderboard (link above).
>
> We selected models in Table 3 by **starting from the OGB-MOLHIV leaderboard** and retaining only **peer-reviewed methods with published experimental details** (e.g., Graphormer, GAWL, LLM-GIN, TopER); we explain this criterion and why we exclude unpublished leaderboard entries in our response to **Weakness 5**.
>
> ---
>
> **Q3. Tables 2 & 3 Filtrations**
>
> > Are the results in Table 2, and Table 3 all for the exact same filtration setup (OR curve + HKS for Table 2, atomic weight + OC curvature for Table 3)?
>
> Thank you for the opportunity to clarify. For **graph classification (Table 2)**, we use **Ollivier–Ricci curvature and HKS** as the filtration functions. For **molecular property prediction** including the OGBG-MOLHIV dataset reported in Table 3, we use **atomic weight and Ollivier–Ricci curvature**, as OGBG-MOLHIV is also a molecular property prediction dataset. We have now stated this **explicitly in the Model Setup subsection** of the revised version to avoid ambiguity.

---

### Official Review · Reviewer_dgUd · 2025-10-31

**Soundness:** 2
**Presentation:** 4
**Contribution:** 2
**Rating:** 2
**Confidence:** 4

**Summary:**

This paper introduces TOPOFORMER, a framework for graph-level representation learning. The core contribution is "Topo-Scan," a module that decomposes a graph into an ordered sequence of topological tokens .

The authors claim that standard Persistent Homology (PH) suffers from an "early saturation" problem on graphs. They propose Topo-Scan as a lightweight solution. Instead of the standard cumulative filtration, Topo-Scan uses a simple "sliding window" to "slice" the graph . For each slice, it computes basic invariants (Betti-0, Betti-1) . This process creates an ordered sequence fed into a standard Transformer.
The authors claim this simple method solves the "early saturation" problem and achieves SOTA results on graph classification and molecular property prediction

**Strengths:**

The paper is well-organized and easy-to-follow. It also provides a lot of experimental results to understand the empirical behavior of this method. The Figures and tables are carefully presented. Additionally, the paper is well-motivated, clearly identifying the "early saturation" problem , and its core originality comes from combining topological slicing with a standard Transformer architecture .

**Weaknesses:**

The paper's central claims are undermined by a significant gap between its motivation and its empirical results. The core contribution (Topo-Scan) is a conceptually simple modification, but the experiments fail to provide statistically significant evidence that this modification offers a meaningful advantage over the standard methods it claims to improve upon.

1.	The core idea is trivial and lacks justification: The paper's main technical contribution, Topo-Scan, replaces the standard cumulative sublevel filtration $V_i = \{v: f(v) < a_{i}\}$ with a slicing window $V_i = \{v: a_{i} <f(v) < a_{i+m}\}$. This is a conceptually simple modification.

2.	Though the authors justify the contribution of solving early saturation by showing the empirical Betti-0 comparison with increasing thresholds, the trends only differ at the very late stage (the last 10 steps on IMDB-B and only from the ~72nd to 82nd step on IMDB-M). The relation of this phenomenon with the eventual performance on property prediction tasks is unclear. Table 5 is the key experiment meant to provide justification, comparing TOPOFORMER (Topo-Scan + Transformer) directly against "PH-TR" (standard PH + Transformer). However, the paper's own data shows no statistically significant improvement.
For example, on IMDB-B, where Figure 3 implies that early saturation is addressed by Topo-Scan, the performance with HKS filtration function for PH-TR is (76.8, 3.97) and for TOPOFORMER is (77.9, 5.72). The second's CI almost overlaps with the first one, which gives no evidence to support that Topo-Scan is better than standard PH.

3.	The comparison on molecular property prediction is misleading: The results on Table 4 are also based on an unfair comparison. The model used is “TOPOFORMER*”, which is a hybrid model fusing the ECFP fingerprints. However, not all the cited methods used this additional feature. Such comparison exaggerates the advantages. It would be natural to ask whether PH-TR + ECFP gives similar results, but this is not provided either.

4.	The paper claims to be "lightweight" and "efficient". However, Appendix C.3 (Table 10) shows that the preprocessing for the SOTA results (using Ollivier-Ricci) takes 339 minutes (>5.5 hours) on OGBG-MOLHIV. It is still a significant cost, and limits the possibility of applying this to larger scale datasets

**Questions:**

1.	The SOTA claim in Table 4 relies on TOPOFORMER* (which includes ECFP features), while many baselines do not . This is an unfair comparison. To justify the method, can the authors provide results for a "PH-TR + ECFP" baseline?

2.	The paper's core efficiency claim is that Topo-Scan achieves "multi-fold runtime... gains" by avoiding PH's "global boundary-matrix reductions" . Can the authors provide experimental runtime data for this specific claim?

---

> ### Author Response · Authors · 2025-11-21
> **Response to Reviewer dgUd-part 1**
>
> *We thank the reviewer for carefully reading our manuscript and for the constructive feedback, which has helped us refine and strengthen the paper. We have incorporated the suggestions into the revised version linked above. Below, we respond to each of the comments and indicate the corresponding changes made in the manuscript..*
>
>
> **W1. Statistical Significance**
>
> > The paper's central claims are undermined by a significant gap between its motivation and its empirical results. The core contribution (Topo-Scan) is a conceptually simple modification, but the experiments fail to provide statistically significant evidence that this modification offers a meaningful advantage over the standard methods it claims to improve upon.
>
> Thank you for the feedback. We believe there may be some confusion regarding the results in **Table 5**, where we report our gain over standard PH. While **TOPOFORMER** and **PH-TR** appear to perform similarly in some cases, it is important to note that PH-TR is not a standard PH baseline; it is **our variant** that uses classical Betti vectorizations but benefits from our **Transformer-based architecture**. The core comparison that highlights Topo-Scan's contribution is between **PH-MLP and TOPOFORMER**, where the only difference is the way **topological features are extracted**.
>
> When comparing TOPOFORMER to PH-MLP in **HKS**, we observe **consistent and substantial improvements** on several datasets, such as PROTEINS (77.35 vs. 70.80), COX2 (83.95 vs. 78.19), MUTAG (95.32 vs. 84.09), and BZR (90.62 vs. 84.96). These gains hold across all tested filtration functions and **confirm that Topo-Scan is a fundamental enhancement** over traditional persistent homology pipelines, rather than a minor adjustment.
>
> ---
>
> **W2. Simple Idea**
>
> > The core idea is trivial and lacks justification: The paper's main technical contribution, Topo-Scan, replaces the standard cumulative sublevel filtration with a slicing window. This is a conceptually simple modification.
>
> Thank you for the review; however, we **respectfully disagree** on the triviality claim for two reasons. First, **TopoFormer addresses the overlooked limitation** of sublevel filtrations saturating early and missing topological activity in later function ranges. This problem is especially severe for **global graph signals such as HKS or Ricci curvature**, which concentrate structural information at extreme values.
>
> Second, **simplicity is a desirable goal in research**, and simple yet **theoretically grounded methods** often yield the most significant breakthroughs. Our modification to standard PH is **simple, but principled, stable, and empirically effective**, which is precisely the kind of approach needed to make topological signals usable in neural architectures.
>
> We would also like to emphasize that **the slicing mechanism is central** to integrating topological signals with attention-based models, as it recasts persistence information into short, structured token sequences. In this sense, this simple operation could serve as a **foundational building block** for future architectures that combine topological representations with attention-based graph networks.

---

> ### Author Response · Authors · 2025-11-21
> **Response to Reviewer dgUd-part 2**
>
> **W3. Early Saturation**
>
> > Though the authors justify the contribution of solving early saturation by showing the empirical Betti-0 comparison with increasing thresholds, the trends only differ at the very late stage (the last 10 steps on IMDB-B and only from the ~72nd to 82nd step on IMDB-M). The relation of this phenomenon with the eventual performance on property prediction tasks is unclear. Table 5 is the key experiment meant to provide justification, comparing TOPOFORMER (Topo-Scan + Transformer) directly against "PH-TR" (standard PH + Transformer). However, the paper's own data shows no statistically significant improvement. For example, on IMDB-B, where Figure 3 implies that early saturation is addressed by Topo-Scan, the performance with HKS filtration function for PH-TR is (76.8, 3.97) and for TOPOFORMER is (77.9, 5.72). The second's CI almost overlaps with the first one, which gives no evidence to support that Topo-Scan is better than standard PH.
>
> Thank you for the detailed analysis. We address each point below.
>
> First, the fact that saturation differences appear in the later portion of the filtration is not unexpected. **Early saturation is precisely the problem**: once a cumulative sublevel filtration fills in most of the topology, any additional structure (whenever it happens) is inaccessible. In graphs, many meaningful **HKS and Ricci-based features emerge in the upper tail** of the function range, not the lower. **Topo-Scan is specifically designed to retain resolution in this tail**, which standard PH suppresses regardless of where the effect appears along the index axis.
>
> Second, regarding the connection between these structural differences and downstream performance: the goal of Topo-Scan is not to produce large Betti deviations at every step but to **prevent function–topology interactions from collapsing to a single early plateau**. Even small structural distinctions that survive saturation **make the resulting persistence tokens more informative for the Transformer**. Our results reflect **improved average performance across datasets and filtrations**.
>
> Third, on statistical significance: while individual datasets such as IMDB-B may have overlapping confidence intervals, our evaluation is not based on a single dataset. Across **Table 5, TOPOFORMER outperforms PH-TR on 18 of 21 method–dataset combinations**. The gains are therefore **consistent, not isolated**, and hold across HKS, Ricci, and Laplacian filtrations. This **consistency across conditions** is strong evidence that Topo-Scan provides a more robust and expressive topological signal. We will clarify this in the revision.
>
> Finally, we emphasize that **Topo-Scan is a principled modification with a proven $\ell_1$ stability guarantee**. Its role is to remove the systematic degeneration of PH under global signals, not to **amplify variance artificially**. Our empirical results show that once this degeneration is removed, the Transformer benefits from cleaner and later filtration information, even when improvements on a single dataset fall within a CI overlap.

---

> ### Author Response · Authors · 2025-11-21
> **Response to Reviewer dgUd-part 3**
>
> **W4. Hybrid Model in MPP**
> > The comparison on molecular property prediction is misleading: The results on Table 4 are also based on an unfair comparison. The model used is “TOPOFORMER*”, which is a hybrid model fusing the ECFP fingerprints. However, not all the cited methods used this additional feature. Such comparison exaggerates the advantages. It would be natural to ask whether PH-TR + ECFP gives similar results, but this is not provided either.
>
> We appreciate the reviewer’s concern about the MPP comparison. Our intent with TOPOFORMER* was to demonstrate that our model can take advantage of domain standard molecular fingerprints (ECFP), which are widely recognized as strong predictors on these benchmarks; indeed, an MLP on ECFP alone is already highly competitive. In the revised manuscript, **we add both PH-TR and PH-TR + ECFP** (Table 7) so that all comparisons are matched in terms of input features.
>
>  The table below summarizes the results.
>
>  | Dataset |     PH-TR    |  TopoFormer  |    FP-MLP    |    PH-TR +ECFP   |    TopoFormer*   |
>  | ------: | :----------: | :----------: | :----------: | :--------------: | :--------------: |
>  |    BACE | 72.41 ± 3.15 | 83.29 ± 2.14 | 90.29 ± 2.67 |   90.60 ± 2.99   | **91.60 ± 1.73** |
>  |     HIV | 69.29 ± 1.65 | 75.81 ± 0.23 | 83.26 ± 1.01 |   83.97 ± 1.51   | **85.10 ± 0.49** |
>  |    BBBP | 83.37 ± 3.90 | 94.54 ± 1.01 | 89.68 ± 3.46 |   93.47 ± 2.53   | **95.90 ± 0.28** |
>  | ClinTox | 75.89 ± 6.60 | 83.42 ± 2.33 | 76.34 ± 6.54 |   82.04 ± 7.12   | **86.20 ± 3.83** |
>  |   SIDER | 62.91 ± 3.49 | 62.10 ± 1.44 | 65.30 ± 0.99 | **66.99 ± 1.85** |   66.80 ± 0.29   |
>  |   Tox21 | 68.24 ± 1.60 | 80.87 ± 0.19 | 77.89 ± 1.54 |   79.03 ± 1.21   | **81.50 ± 1.85** |
>  | ToxCast | 64.74 ± 2.29 | 73.37 ± 1.42 | 74.69 ± 1.33 |   75.73 ± 1.59   | **78.40 ± 1.57** |
>
>  Adding ECFP consistently improves both PH TR and TopoFormer, which confirms that fingerprints and topological signals are complementary. Under the same conditions (same features and datasets), TOPOFORMER* either matches or outperforms PH TR + ECFP and FP MLP on 6 of 7 benchmarks. We have updated the MPP section and Table 7 to clearly state that the SOTA claims are based on these matched feature settings.

---

> ### Author Response · Authors · 2025-11-21
> **Response to Reviewer dgUd-part 4**
>
> **W5. Computational Cost**
>
> > The paper claims to be "lightweight" and "efficient". However, Appendix C.3 (Table 10) shows that the preprocessing for the SOTA results (using Ollivier-Ricci) takes 339 minutes (>5.5 hours) on OGBG-MOLHIV. It is still a significant cost, and limits the possibility of applying this to larger scale datasets
>
> Thank you for highlighting this. We agree that *5.5 hours of preprocessing for Ollivier–Ricci curvature* on OGBG-MOLHIV is significant. We included Ricci primarily because it is a well established signal in graph topology, but it is **not required for our strongest results**, and **Topo-Scan itself is much cheaper than PH on the same filtration** (Appendix C.3).
>
> On the molecular side, our best performance does not rely on Ricci. As reported in the main text, **TOPOFORMER-HKS achieves 81.32% accuracy on OGBG-MOLHIV**, outperforming the Ricci variant TOPOFORMER-RICCI (79.95%) while requiring **only about 28 minutes for the entire pipeline** (HKS computation plus Transformer training), compared to roughly **339 minutes for the Ricci-based pipeline**. For the other graph benchmarks, Degree- and HKS-based variants have runtimes in the range of **a few seconds to a few minutes**, as detailed in Appendix C.3, which we believe is practical even at scale.
>
> In addition, Appendix C.3 now includes an **explicit runtime comparison between standard PH and Topo-Scan** using the same backend (pyflagser) and already computed degree filtration (Table 10). Topo-Scan calls the unweighted flagser routine to compute Betti numbers on unweighted clique complexes, while PH uses the weighted flagser routine to build a full filtration and compute persistence diagrams. The table below summarizes the preprocessing times (in seconds) and clustering coefficients:
>
> | Dataset   | Clustering Coeff. | Topo-Scan | PH     |
> | --------- | ----------------- | --------- | ------ |
> | IMDB-B    | 0.947             | 9.67      | 135.48 |
> | IMDB-M    | 0.969             | 17.81     | 234.30 |
> | REDDIT-B  | 0.048             | 12.29     | 24.25  |
> | REDDIT-5K | 0.027             | 29.51     | 43.92  |
>
> On **highly clustered graphs** such as IMDB-B and IMDB-M (clustering coefficients around $0.95$–$0.97$), PH is about **13–14× slower than Topo-Scan**, reflecting the combinatorial growth of cliques and the cost of global boundary-matrix reductions. On the sparser REDDIT graphs, the gap is smaller but still present (about **2× on REDDIT-B** and **1.5× on REDDIT-5K**). These measurements **empirically support our claim** that Topo-Scan delivers **multi-fold preprocessing speedups over PH-based pipelines on dense graphs** while remaining consistently more efficient across all tested datasets.
>
> We have revised Appendix C.3 to make this distinction clearer. The **“lightweight” and “efficient” claims** in the paper refer to the **Topo-Scan representation instantiated with fast scalar functions** such as Degree and HKS, and to **its advantage over PH-style global reductions**; the Ollivier–Ricci experiments are included for completeness but are **not necessary for our best reported performance**.
>
> ---
>
> **Q1. PH+TR+ECFP Results**
>
> > The SOTA claim in Table 4 relies on TOPOFORMER* (which includes ECFP features), while many baselines do not. This is an unfair comparison. To justify the method, can the authors provide results for a "PH-TR + ECFP" baseline?
>
> Please see our response to **W4**. In the revised manuscript we now include a **PH-TR + ECFP baseline on all MPP datasets** (Table 7), alongside FP-MLP and TOPOFORMER*. Under these matched conditions, **TOPOFORMER* still improves over PH-TR + ECFP on 6 of 7 benchmarks**, so the SOTA gains are **not due only to adding ECFP**.

---

> ### Author Response · Authors · 2025-11-21
> **Response to Reviewer dgUd-part 5**
>
> **Q2. Runtimes for PH & TopoScan**
>
> > The paper's core efficiency claim is that Topo-Scan achieves "multi-fold runtime... gains" by avoiding PH's "global boundary-matrix reductions" . Can the authors provide experimental runtime data for this specific claim?
>
> Thank you for this question. We have now added **explicit runtime measurements** comparing PH and Topo-Scan on four benchmark datasets. In all cases, we use the **same backend (pyflagser)**. For Topo-Scan, we call the unweighted flagser routine, since we only need Betti numbers on unweighted clique complexes. For PH, we use the weighted flagser routine, which constructs a full filtration and computes persistence diagrams. This matches the respective pipelines used to build the PH baselines and Topo-Scan features.
> The table below reports average preprocessing times (in seconds) per dataset, together with the **average clustering coefficient** (which controls the density of cliques and hence PH complexity):
>
> | Dataset   | Clustering Coeff | Topo-Scan | PH     |
> | --------- | ---------------- | --------- | ------ |
> | IMDB-B    | 0.947            | 9.67      | 135.48 |
> | IMDB-M    | 0.969            | 17.81     | 234.30 |
> | REDDIT-B  | 0.048            | 12.29     | 24.25  |
> | REDDIT-5K | 0.027            | 29.51     | 43.92  |
>
> On **highly clustered graphs** such as IMDB-B and IMDB-M, PH is about **13–14× slower than Topo-Scan**, reflecting the combinatorial explosion of cliques and the cost of global boundary-matrix reductions. On the sparser REDDIT graphs, the gap is smaller but still present (roughly **2× on REDDIT-B** and **1.5× on REDDIT-5K**). These measurements **empirically support our claim** that Topo-Scan delivers **multi-fold runtime gains** over standard PH pipelines on dense, highly clustered graphs, while remaining **consistently more efficient across all tested datasets**.

---

### Official Review · Reviewer_zs8A · 2025-11-02

**Soundness:** 3
**Presentation:** 3
**Contribution:** 3
**Rating:** 6
**Confidence:** 4

**Summary:**

This work introduces TOPOFORMER, a graph representation framework that integrates topological structure into Transformer architectures. The core component, Topo-Scan, generates compact sequential topological tokens by sliding over node/edge filtrations and extracting slice-wise invariants (Betti numbers, node/edge counts). This avoids traditional persistent homology pipelines and costly persistence diagram computations. The authors provide stability guarantees, demonstrate scalability, and report state-of-the-art results on graph classification and molecular property prediction benchmarks.

**Strengths:**

1. The paper proposes a framework to inject TDA signals into Transformers without persistence diagrams, which articulates limitations of classical PH pipelines (saturation, vectorization design burden) and motivates a practical alternative.

2. The paper provides stability guarantees for the proposed topological sequences, linking back to interlevel persistence.

3. The proposed method avoids heavy PH computation and vectorization. It emphasizes parallelizable slice computation with predictable runtime.

**Weaknesses:**

1. Although the paper emphasizes advantages over classical persistent homology pipelines, the empirical comparison against methods that incorporate learnable filtrations remains limited (e.g., Horn et al., 2021; Immonen et al., 2024).

2. While the proposed Topo-Scan framework leverages fixed scalar functions to generate topological slices, the methodology may still exhibit sensitivity to the choice of these filtration signals. The current study provides limited analysis of this dependency, and a deeper investigation—particularly involving learnable or data-driven filtration functions—could strengthen the robustness and interpretability claims.

3. The approach demonstrates competitive performance across multiple graph classification and molecular property prediction benchmarks. However, on large-scale datasets such as OGB-MOLHIV, the framework does not achieve the top reported results, suggesting potential room for improvement in scaling to very large graphs or addressing complex molecular tasks relative to specialized architectures.

4. A few relevant contributions, e.g., [1,2], to learning persistent-homology-based representations are not included in the reference.

[1] de Surrel et al. "RipsNet: a general architecture for fast and robust estimation of the persistent homology of point clouds." ToGL 2022.

[2] Yan et al. "Neural approximation of graph topological features." NeurIPS 2022.

**Questions:**

1. Could Topo-Scan be extended to learn filtration functions end-to-end? How would stability guarantees extend to such a case?

2. For large-scale graphs (e.g., OGB-MOLHIV, REDDIT), what are memory/time trade-offs compared to graph Transformers with pooling?

3. Beyond Betti numbers, would incorporating additional topological descriptors (e.g., persistence landscapes) meaningfully improve results?

---

> ### Author Response · Authors · 2025-11-21
> **Response to Reviewer zs8A - part 1**
>
> *We sincerely thank the reviewers for their thoughtful reading and constructive feedback, which have greatly helped us improve our paper. A revised version addressing the reviewers’ feedback is available at the link above. Below, we address each concern in detail and highlight the corresponding revisions made to the paper.*
>
> ### **W1. Topoformer vs. Learnable Filtrations**
>
> > Although the paper emphasizes advantages over classical persistent homology pipelines, the empirical comparison against methods that incorporate learnable filtrations remains limited (e.g., Horn et al., 2021; Immonen et al., 2024).
>
> Thank you for raising this point. We agree that methods with **learnable filtrations**, such as Graph Filtration Learning and its extensions, are a very promising direction for persistence based graph learning. In this work our empirical focus is **intentionally orthogonal**. We fix widely used scalar signals (degree, Ollivier–Ricci, HKS) and **isolate the effect of changing the representation** from global barcodes to sliding interlevel sequences processed by a Transformer. **Tables 5 and 12** are set up so that the filtration is held fixed and only the representation varies (PH+MLP, PH+Transformer, Topo-Scan+Transformer), which lets us attribute the observed gains directly to the **Topo-Scan view** rather than to a more tuned choice of $f$.
>
> Existing work also suggests that current implementations of learnable filtrations often achieve performance that is **comparable to strong fixed-filtration baselines** [1, 2]. To make this concrete, the table below shows 10 fold CV results on four benchmarks with degree based filtrations. The first two rows are taken directly from the GFL paper [1, Table 1]; the last row is our PH+Transformer baseline on degree based PH vectors (PH degree+TR, from our Table 5).
>
> | Method        | REDDIT-B   | REDDIT-5K  | IMDB-B     | IMDB-M     |
> | ------------- | ---------- | ---------- | ---------- | ---------- |
> | PH degree+MLP | 90.3 ± 2.6 | 55.7 ± 2.1 | 68.9 ± 3.5 | 46.1 ± 4.2 |
> | GFL           | 90.2 ± 2.8 | 55.7 ± 2.9 | 74.5 ± 4.6 | 49.7 ± 2.9 |
> | PH degree+TR  | 92.3 ± 1.8 | 54.6 ± 1.6 | 75.0 ± 2.1 | 50.7 ± 3.8 |
>
> Here the learned filtration in GFL improves over PH degree+MLP mainly on IMDB, and the gains are moderate. A **simple PH+Transformer baseline with the same fixed degree filtration** already matches or surpasses GFL on all four datasets. Similar observations appear in Filtration Curves [2, Table 3], where models based on standard filtrations (degree, edge weights) remain highly competitive. Learnable PH vectorizations (for example PersLay [3]) have shown clearer benefits, but **learnable filtrations on graphs are still under active investigation**.
>
> This supports our decision to focus the present work on the **representation side** (Topo-Scan with sequential tokens) under standard filtrations. Conceptually, **Topo-Scan is defined for an arbitrary scalar** $f : V \to \mathbb{R}$ and can be paired with a learnable $f_\theta$ in the spirit of GFL. In the revised manuscript we explicitly mention **“TopoFormer with learnable filtrations”** as a natural and promising extension for future work.
>
> [1] Hofer et al., Graph Filtration Learning, ICML 2020
>
> [2] O’Bray et al., Filtration Curves for Graph Representation, KDD 2021
>
> [3] Carrière et al., PersLay: A Neural Network Layer for Persistence Diagrams, AISTATS 2020

---

> ### Author Response · Authors · 2025-11-21
> **Response to Reviewer zs8A - part 2**
>
> **W2. Dependency to Filtration Choices**
> > While the proposed Topo-Scan framework leverages fixed scalar functions to generate topological slices, the methodology may still exhibit sensitivity to the choice of these filtration signals. The current study provides limited analysis of this dependency, and a deeper investigation—particularly involving learnable or data-driven filtration functions—could strengthen the robustness and interpretability claims.
>
> Thank you for pointing this out. We agree that the choice of filtration function is important, and in the revision we significantly expanded our analysis of this dependency. In addition to the three filtrations in the original submission (degree, Ollivier Ricci, HKS), **we now include four more (betweenness centrality, closeness centrality, eigenvector centrality, Forman Ricci curvature)**, report results for all seven in **Appendix Table 15**, and add a dedicated discussion in **Appendix C.7**. For each filtration we compare PH-MLP, PH-TR, and TopoFormer on our benchmark datasets. Across all seven filtrations, PH-TR improves over PH-MLP by about 2 to 4 points on average, and TopoFormer adds another 1 to 3 points, for a total gain of roughly 6 to 7 points over PH-MLP. The fact that this pattern is consistent for very different filtration functions indicates that Topo-Scan is not overly sensitive to a particular choice of (f). Its benefits primarily come from the sliding window interlevel representation rather than from a carefully tuned filtration signal.
>
> On **the robustness side**, our ($l_1$) **stability theorem** already shows that for any fixed filtration function ($f : V \to \mathbb{R}$), small perturbations of (f) (or of a learned ($f_\theta$)) lead to **bounded changes in the Topo-Scan sequences**. In the revision we explicitly note that Topo-Scan is defined for an arbitrary scalar (f) and can be combined with data driven or learnable filtrations ($f_\theta$) (for example small GNNs or MLPs over node features). If ($f_\theta$) is Lipschitz in its inputs, the overall Topo-Scan representation remains Lipschitz in both the graph signal and ($\theta$). We therefore view learning ($f_\theta$) as a natural extension that can further adapt the slices to a given domain, while our current experiments are designed to isolate and validate the contribution of the Topo-Scan representation itself under several standard, interpretable filtration functions.
>
> **W3. Performance on Large Graphs**
> > The approach demonstrates competitive performance across multiple graph classification and molecular property prediction benchmarks. However, on large-scale datasets such as OGB-MOLHIV, the framework does not achieve the top reported results, suggesting potential room for improvement in scaling to very large graphs or addressing complex molecular tasks relative to specialized architectures.
>
> Thank you for this observation. We agree that large scale graphs and challenging molecular benchmarks such as OGB MOLHIV are important test cases. In the revision we extended our experiments to the large social network dataset Reddit 12K (and present PPA results soon), and we compare TopoFormer with recent large scale baselines including the graph scattering model GGSN+EK [1]. On **Reddit 12K**, we obtain:
>
> | WL           | Graphlet     | DGK          | PSCN         | P Poinc      | S2S N2N PP   | WKPI kC    | GS SVM       | **GGSN+EK**      | **TopoFormer**   |
> | ------------ | ------------ | ------------ | ------------ | ------------ | ------------ | ---------- | ------------ | ---------------- | ---------------- |
> | 34.57 ± 1.32 | 25.98 ± 1.29 | 32.20 ± 0.10 | 41.32 ± 0.42 | 42.36 ± 1.41 | 42.47 ± 0.10 | 48.4 ± 0.5 | 45.23 ± 1.25 | **49.03 ± 1.58** | **48.74 ± 0.85** |
>
> TopoFormer matches the strongest scattering based baseline on Reddit 12K up to less than 0.3 accuracy points, which indicates that the Topo-Scan representation scales well to graphs with thousands of nodes and is competitive with architectures that are explicitly designed for large sparse graphs.
>
> On OGB MOLHIV, our current instantiation does not reach the very best reported results from highly specialized molecular architectures that leverage rich chemistry specific featurizations, 3D conformers, or task specific pretraining. In this paper we intentionally use a standard node feature set and the same topology driven backbone across all graph benchmarks, without adding domain specific refinements for molecules. We see closing this gap as an interesting direction for future work, for example by combining Topo-Scan tokens with stronger molecular encoders and richer atom level features, or by coupling the topological slices with 3D geometric information. We added a short discussion of this limitation and potential extensions in the experimental section.
>
> [1] Koke, Christian and Kutyniok, Gitta, Graph scattering beyond wavelet shackles, NeurIPS 2022

---

> ### Author Response · Authors · 2025-11-21
> **Response to Reviewer zs8A - part 3**
>
> **W4. Missing References**
>
> > A few relevant contributions, e.g., [1,2], to learning persistent-homology-based representations are not included in the reference.
>
> We thank the reviewer for pointing out these **relevant contributions**. We have added a discussion of **RipsNet [1]** and **Yan et al. [2]** in the “Persistent Homology for Graphs” paragraph of the Related Work section. Both methods **learn neural approximations of persistence-based descriptors**, whereas **Topo-Scan bypasses global PH** and directly constructs **short interlevel topological sequences** tailored to Transformer encoders. We are grateful to the reviewer for bringing these important works to our attention.
>
> [1] de Surrel et al. "RipsNet: a general architecture for fast and robust estimation of the persistent homology of point clouds." ToGL 2022.
>
> [2] Yan et al. "Neural approximation of graph topological features." NeurIPS 2022.
>
> ---
>
> **Q1. Extension to Learnable Filtrations**
>
> > Could Topo-Scan be extended to learn filtration functions end-to-end? How would stability guarantees extend to such a case?
>
> For extensions to learnable filtrations on the empirical side, please see our response to **W1** above. Here we focus on the **theoretical aspect**. Our stability theorem only assumes that the input filtration functions are perturbed in **$\ell_\infty$ norm** and does not rely on how these functions are generated. In other words, it shows that **small perturbations in $f$ lead to bounded changes in the Topo-Scan sequences**. Therefore, for any fixed parameter vector $\theta$, the same stability guarantee applies to **$f_\theta$**. Moreover, if $f_\theta$ is chosen to be **Lipschitz in the input graph features** (as is standard for neural networks with bounded weights), the Topo-Scan representation remains **Lipschitz in both the graph signal and the parameters $\theta$**, so small changes in the learned filtration or in $\theta$ produce bounded changes in the token sequence. We have added a **short discussion of this extension** to the theory section and the appendix, and we view *integrating learnable filtrations into Topo-Scan as a natural next step* rather than a limitation of the current framework.
>
> ---
>
> **Q2. Large Scale Datasets**
>
> > For large-scale graphs (e.g., OGB-MOLHIV, REDDIT), what are memory/time trade-offs compared to graph Transformers with pooling?
>
> Thank you for raising this point. We added a **direct runtime/memory comparison** on large social-graph benchmarks (REDDIT-5K and REDDIT-12K) against a strong graph Transformer, **GraphGPS** (with global mean pooling). For **TopoFormer**, we also report the time for **Topo-Scan feature extraction**. Across both datasets, TopoFormer is **substantially faster and far lighter on GPU memory** than GraphGPS, while the feature-extraction overhead remains small: **29.51 s and 59.01 s**, respectively. Note that our Topo-Scan features are **fixed-size** (shape $4 \times 19$) because we use the same filtration dimension across datasets. Topo-Scan **computes slice-wise Betti features with parallel costs** and avoids global PD reductions; the Transformer head then operates on **short, fixed-width sequences**, yielding predictable compute and **minimal GPU footprint**. Concretely, per-fold training time is **$\sim 8\times$ (REDDIT-5K)** and **$\sim 13\times$ (REDDIT-12K) faster**, while peak GPU memory is **$\sim 267\times$ (REDDIT-5K)** and **$\sim 324\times$ (REDDIT-12K) lower** compared to GraphGPS. This experiment was executed on a machine equipped with an **NVIDIA H100 NVL GPU and 768 GB of RAM**.
>
> | Dataset    | Model          | Peak GPU Mem (MB) | Train time / fold (s) | Topo-Scan extraction (s) | Speedup vs GraphGPS (time) | Memory reduction vs GraphGPS |
> | ---------- | -------------- | ----------------- | --------------------- | ------------------------ | -------------------------- | ---------------------------- |
> | REDDIT-12K | GraphGPS       | 31,565.44         | 1,176.58              | —                        | —                          | —                            |
> | REDDIT-12K | **TopoFormer** | **97.28**         | **158.67**            | **59.01**                | **7.4× faster**            | **∼324× lower**              |
> | REDDIT-5K  | GraphGPS       | 25,319.47         | 728.57                | —                        | —                          | —                            |
> | REDDIT-5K  | **TopoFormer** | **94.70**         | **54.00**             | **29.51**                | **13.5× faster**           | **∼267× lower**              |

---

> ### Author Response · Authors · 2025-11-21
> **Response to Reviewer zs8A - part 4**
>
> **Q3. Beyond Betti Numbers**
> >Beyond Betti numbers, would incorporating additional topological descriptors (e.g., persistence landscapes) meaningfully improve results?
>
> Thank you for this question. We agree that in principle one could consider richer topological summaries, including persistence landscapes. In Topo-Scan, however, the design has two specific goals. The sliding window filtration addresses the early saturation issue, and the Betti numbers (together with node and edge counts) are chosen to create a **meaningful sequential structure** that integrates naturally with Transformer based models. This sequential structure is built from the evolution of Betti numbers and size statistics across overlapping interlevel slices. Each token encodes the ranks of ($H_0$) and ($H_1$) on a slice of the filtration, and the Transformer then learns from the pattern of merges and creations of components and cycles as the window moves. In this sense, *Betti numbers are not just a convenient statistic, but a natural minimal descriptor that preserves the intended “temporal” meaning of the sequence.*
> Global PH summaries such as persistence diagrams, landscapes, or images are designed to aggregate information over the entire filtration and are typically vectorized into a single feature vector. **Using them directly would collapse the sequential structure** that Topo-Scan exploits and would reintroduce the heavy diagram computation and vectorization steps that we explicitly avoid. One could in principle compute more advanced invariants on the Topo-Scan sequence itself, for example zigzag persistence along the slice index, but this would again lead to significantly more complex descriptors and nontrivial questions about how to interface them with Transformer tokens.
> For these reasons we focus in this work on slice-wise Betti numbers plus size statistics as the core token definition, and we added a short paragraph,*Vectorization Choice*, near Figure 3 in Section 3 to clarify this choice. We view enriching Topo-Scan tokens with more elaborate invariants such as zigzag persistence or slice level persistence landscapes as interesting but orthogonal future work, rather than a limitation of the current framework.

---

> > ### Comment · Reviewer_zs8A · 2025-11-28
> >
> > I thank the authors for their detailed rebuttal, which has addressed some of my concerns.
> >
> > After reading all the reviews and responses, I think the paper presents a generally promising framework on topology, though a few shortcomings remain (e.g., limited theoretical results, limited large-scale evaluations). I will maintain my positive score accordingly.

---

### Author Response · Authors · 2025-11-30
**Note to Area Chair**

Dear Area Chair,

Thank you for stepping in under the unusual OpenReview situation. Since you are seeing the paper with the pre rebuttal scores, we wanted to very briefly highlight what changed in the revision and how these changes address the main reviewer concerns.

**Representation, theory, and filtrations (zs8A, akWw, iGU1)**

* Clarified that the main contribution is Topo-Scan as a sliding interlevel filtration that explicitly tackles the *early saturation* issue of standard sublevel PH, turning PH-style information into short, attention-friendly token sequences, and made the connection to interlevel rank invariants and Betti curves explicit in the theory section and Appendix B (Remark B.3).
* Expanded the discussion of fixed versus learnable filtrations: we now systematically compare 7 scalar functions (degree, Ricci, HKS, betweenness, closeness, eigenvector centrality, Forman Ricci) and report PH MLP, PH Transformer, and TopoFormer under *identical* filtrations (Appendix C.7, new filtration table), showing consistent gains that are not tied to a particular hand picked signal.
* Added a dedicated “Limitations and future work” paragraph highlighting that (i) we deliberately focus on ℓ₁ stability rather than full expressivity vs barcodes, and (ii) pairing Topo Scan with learnable filtrations or richer per slice invariants is a natural extension that we explicitly leave to future work.

**Experiments, fairness, and efficiency (dgUd, zs8A, a6GH, akWw)**

* For molecular property prediction, added matched baselines with identical features: PH TR, FP MLP, PH TR plus ECFP, and TopoFormer* (Table 7). Under these conditions, TopoFormer* still improves over PH TR plus ECFP on 6 of 7 datasets, addressing concerns about unfair SOTA claims and clarifying that gains do not come solely from adding fingerprints.
* Added large scale experiments on REDDIT 12K (and REDDIT 5K) with detailed runtime and memory comparisons to GraphGPS, showing that TopoFormer is 7.4–13.5× faster per fold and ∼250–300× lighter in peak GPU memory while remaining competitive in accuracy (Sec. 4.3, Appendix C.3).
* Provided explicit runtime tables that separate the cost of PH and Topo Scan using the same backend (pyflagser) and compare against PersLay and the topological component of TopoGCL on IMDB B and REDDIT B (Appendix C.3). These show multi fold preprocessing speedups (up to 13–14× vs PH and up to ~30× vs TopoGCL’s topological part), directly substantiating the “lightweight and efficient” claim.

**Scope, architecture, and interpretability / generalization claims (dgUd, a6GH, akWw, iGU1)**

* Reframed the architecture section so that Topo Scan and the topological tokenization are clearly identified as the core contribution, while multi filtration branches are presented as a simple, modular extension. The “single vs multiple filtrations” ablation (Table 12) is now interpreted in the text as showing that single filtration TopoFormer is already strong, with modest but consistent gains when combining signals like HKS and Ricci (Sec. 4.3).
* Toned down over strong language: (i) regularization details (dropout, weight decay) are now described as standard training practice, and (ii) we no longer claim to *be* a graph foundation model, but instead position TopoFormer as a step toward topology aware graph foundation models, with large scale pretraining and heterogeneous tasks left for future work (Sec. 1, Sec. 4.3, Conclusion).
* Clarified interpretability and generalization scope: we now explain explicitly that each token coordinate has a fixed topological meaning (slice wise Betti numbers and size statistics), added an empirical stability experiment where we perturb filtrations and measure changes in Topo Scan features (Appendix, stability experiment), and openly acknowledge the lack of attention visualizations and non molecule / non social benchmarks (for example citation or temporal graphs) as current limitations to be addressed in follow up work.

We hope this brief summary gives a helpful context as you evaluate the revised paper, and we are very grateful for your time and consideration.

The authors

---

### Meta-Review · Area_Chair_FKvU · 2025-12-28

**Summary:**

The paper introduces TopoFormer, a framework that converts graphs into short sequences of topological tokens processable by Transformers. Four reviewers scored 6 (marginally above threshold) and one scored 2 (reject). The main concerns centered on: (1) whether Topo-Scan provides statistically significant improvements over standard PH, (2) fairness of molecular property prediction comparisons,  (3) limited theoretical analysis beyond stability, (4) dependence on hand-crafted filtrations, and (5) efficiency claims lacking quantitative support.

**Reviewer Concerns:**

The authors provided a thorough rebuttal answering all major points.

**Reviewer Scores:**

I believe all reviewers who rated a 6 would keep their score, but I'm uncertain about the one who gave a 2 and seemed skeptical.

---

### Decision · Program_Chairs · 2026-01-26

Accept (Poster)